


# Validation of Aura-OMI QA4ECV NO₂ Climate Data Records with ground-based DOAS networks: role of measurement and comparison uncertainties

Steven Compernolle[1], Tijl Verhoelst[1], Gaia Pinardi[1], José Granville[1], Daan Hubert[1], Arno Keppens[1], Sander Niemeijer[2], Bruno Rino[2], Alkis Bais[3], Steffen Beirle[4], Folkert Boersma[5,8], John P. Burrows[6], Isabelle De Smedt[1], Henk Eskes[5], Florence Goutail[7], François Hendrick[1], Alba Lorente[8], Andrea Pazmino[7], Ankie Piters[5], Enno Peters[6], Jean-Pierre Pommereau[7], Julia Remmers[4], Andreas Richter[6], Jos van Geffen[5], Michel Van Roozendael[1], Thomas Wagner[4], and Jean-Christopher Lambert[1]

[1]Royal Belgian Institute for Space Aeronomy (BIRA-IASB), Uccle, Belgium
[2]s[&]t Corporation, Delft, The Netherlands
[3]Aristotle University of Thessaloniki, Laboratory of Atmospheric Physics (AUTH), Thessaloniki, Greece
[4]Max Planck Institute for Chemistry (MPIC), Mainz, Germany
[5]Royal Netherlands Meteorological Institute (KNMI), De Bilt, The Netherlands
[6]Institute of Environmental Physics, University of Bremen (IUP-B), Bremen, Germany
[7]Laboratoire Atmosphères, Milieux, Observations Spatiales, CNRS, Guyancourt, France
[8]Wageningen University, Meteorology and Air Quality Group, Wageningen, the Netherlands

**Correspondence:** steven.compernolle@aeronomie.be

**Abstract.** The QA4ECV version 1.1 stratospheric and tropospheric NO₂ vertical column density (VCD) climate data records (CDR) from the satellite sensor OMI are validated, using NDACC zenith scattered light DOAS (ZSL-DOAS) and Multi Axis-DOAS (MAX-DOAS) data as a reference. The QA4ECV OMI stratospheric VCD have a small bias of ~0.2 Pmoleccm⁻² (5-10%) and a dispersion of 0.2 to 1 Pmoleccm⁻² with respect to the ZSL-DOAS measurements. QA4ECV tropospheric
VCD observations from OMI are restricted to near-cloud-free scenes, leading to a negative sampling bias (with respect to the unrestricted scene ensemble) of a few Pmoleccm⁻² up to −10 Pmoleccm⁻² (-40%) in one extreme high-pollution case. QA4ECV OMI tropospheric VCD has a negative bias with respect to the MAX-DOAS data (−1 to −4 Pmoleccm⁻²), a feature also found for the OMI OMNO2 standard data product. The tropospheric VCD discrepancies between satellite and ground-based data exceed by far the combined measurement uncertainties. Depending on the site, part of the discrepancy can be
attributed to a combination of comparison errors (notably horizontal smoothing difference error), measurement/retrieval errors related to clouds and aerosols, and to the difference in vertical smoothing and a priori profile assumptions.

## 1  Introduction

Nitrogen oxides (NO$_x$ = NO₂+NO) play a significant role in the atmosphere, since they catalyse tropospheric ozone formation through a suite of chemical reactions, impact the oxidizing capacity of the atmosphere and thus influence the atmospheric
burdens of major pollutants like methane and carbon monoxide (Seinfeld and Pandis, 1997). In addition, they are responsible





for secondary aerosol formation (Sillman et al., 1990). Fossil fuel combustion is the dominant source to the global NOx emission budget (~50%), followed by natural emissions from soils, lightning and open vegetation fires (Delmas et al., 1997). High ozone, aerosol and $NO_x$ have adverse effects on human health (Hoek et al., 2013; World Health Organization, 2013), and recommended limits from the EU and the World Health Organization are often exceeded especially in densely populated and

industrialized regions (European Environment Agency, 2018). The emissions of $NO_x$ have been therefore the main target of abatement strategies throughout the globe (e.g., the Protocol of Gothenburg, 1999). The effects of $NO_x$ emissions on climate are complex and not fully understood so far. On the one hand, the emissions of $NO_x$ result in the increase of ozone and thus to a net warming (since ozone is a greenhouse gas). On the other hand, they lead to a decrease of methane abundances at longer time scales, and therefore to a cooling effect (Myhre et al., 2013). Due to their indirect impact on radiative forcing and

potential role on climate (Shindell et al., 2009), $NO_x$ are identified as an Essential Climate Variable (ECV) precursor by the Global Climate Observing System (GCOS) (GCOS, 2016). $NO_x$ are also present in the stratosphere (Noxon, 1979), where they contribute to the catalytic destruction of ozone (Crutzen, 1970).

Observations from satellite nadir-viewing sensors are essential for mapping the global multi-year picture of the $NO_x$ distribution and trend. However, the quality of these datasets needs to be carefully assessed, using ground-based measurements

at different sites (see e.g., Petritoli et al., 2004; Pinardi et al., 2014; Heue et al., 2005; Brinksma et al., 2008; Celarier et al., 2008, for validations on GOME, GOME-2. SCIAMACHY and OMI data). A limitation often encountered is that uncertainties in satellite and/or ground-based data are not adequately characterized, and the ground-based datasets are generally not harmonized across networks.

The EU Seventh Framework Programme (FP7) QA4ECV (Quality Assurance for Essential Climate Variables) project

(www.qa4ecv.eu) demonstrated how reliable and traceable quality information can be provided for satellite and ground-based measurements of climate and air quality parameters. We highlight here three of its achievements. (i) The development of a quality assurance framework for climate data records (CDRs) (Nightingale et al., 2018), covering aspects as product traceability, uncertainty description, validation and documentation, following international standards (QA4EO; Joint Committee for Guides in Metrology, 2008, 2012). Among its components are a generic validation protocol (Compernolle et al. (2018), build-

ing further on Keppens et al. (2015)), a compilation of recommended terminology for CDR quality assessment (Compernolle and Lambert, 2017; Compernolle et al., 2018) and a validation server (Compernolle et al., 2016; Rino et al., 2017), the latter being prototype for the operational validation servers for S5P-MPC (Sentinel-5p Mission Performance Center) and CAMS (Copernicus Atmosphere Monitoring Service). (ii) The establishment of multi-decadal CDRs for 6 ECVs along the guidelines of the quality assurance framework. Among them are the QA4ECV $NO_2$ (Lorente et al., 2017; Zara et al., 2018; Boersma et al.,

2018) and HCHO (De Smedt et al., 2018) version 1.1 satellite products, available for several sensors. (iii) The development of a $NO_2$ and HCHO long-term ground-based data set for 10 MAX-DOAS instruments, harmonized in measurement protocol, data format and with extensive uncertainty characterization (Hendrick et al., 2016; Richter et al., 2016).

A general across-community issue in the geophysical validation of satellite data sets with respect to ground-based reference measurements are the additional uncertainties that appear when comparing data sets characterized with different tempo-

ral/spatial/vertical sampling and smoothing properties (Loew et al., 2017). This is especially critical in the case of short-lived





tropospheric gases (Richter et al., 2013b). This issue was the focus of the EU H2020 project GAIA-CLIM (Gap Analysis for Integrated Atmospheric ECV CLImate Monitoring (Verhoelst et al., 2015; Verhoelst and Lambert, 2016).

In this work we report a comprehensive validation of the QA4ECV $NO_2$ version 1.1 data product on the OMI sensor, using as a reference the ground-based measurements acquired by networks of DOAS UV-visible instruments developed in the

context of the Network for the Detection of Atmospheric Composition Change (NDACC). Zenith-scattered light DOAS (ZSL-DOAS) data obtained routinely as part of NDACC monitoring activities is used to validate the stratospheric vertical column density (VCD), while Multi-axis DOAS (MAX-DOAS) data, either from NDACC or further harmonized within the QA4ECV project is used to validate the tropospheric VCD. We focus on how well the ex-ante[1] uncertainties and comparison errors can explain the observed discrepancies, making use of the framework and methodology developed within the projects QA4ECV

and GAIA-CLIM.

In section 2 the satellite and reference data sets are described. Section 3.1 provides details about the validation methodology. In section 3.2 we outline how the quality screening of QA4ECV OMI $NO_2$, notably the exclusion of cloudy scenes, leads to underestimated early afternoon tropospheric $NO_2$ VCDs. Section 3.3 presents the comparison of QA4ECV OMI stratospheric $NO_2$ VCD with ZSL-DOAS. In section 3.4 the satellite tropospheric VCD is compared with measurements from 10 MAX-

DOAS instruments. The differences are analysed in relation to the uncertainties and comparison errors. Potential causes of the discrepancies (e.g., horizontal smoothing difference error, low-lying clouds or aerosols, profile shape uncertainty, etc.) and attempts for resolving the discrepancies are discussed. Finally, the conclusions are formulated in section 4.

---

[1]An ex-ante quantity does not rely on a statistical comparison with external data (von Clarmann, 2006). This is to be contrasted with ex-post quantities like the mean difference of satellite data vs. reference data.





## 2 Description of the data sets

### 2.1 Satellite data

#### 2.1.1 QA4ECV OMI NO$_2$

The QA4ECV NO$_2$ OMI version 1.1 data product is retrieved from level-1 UV-Vis spectral measurements (OMI-Aura_L1-

OML1BRVG radiance files) from the Dutch-Finnish UV-Vis nadir viewing spectrometer OMI (Ozone Monitoring Instrument) on NASA's EOS-Aura polar satellite. The nominal footprint of the OMI ground pixels is 24×13 km $^2$ (across × along track) at nadir to 165×13 km$^2$ at the edges of the 2600 km swath, and the ascending node local time is 13:42 hrs. For more details on the instrument, see Levelt et al. (2006). The data product provides a level-2 (L2) tropospheric, stratospheric, and total NO$_2$ VCD.

The QA4ECV algorithm includes the following steps: (i) the retrieval of the total slant column density (SCD) $N_s$ using Differential Optical Absorption Spectroscopy (DOAS), (ii) estimation of the stratospheric SCD $N_{s,\text{strat}}$ from data assimilation using the chemistry transport model (CTM) TM5, after which (iii) the tropospheric contribution is obtained by subtraction, and (iv) the calculation of tropospheric air mass factors (AMFs) $M_{\text{trop}}$ converting the SCD to a VCD $N_{v,\text{trop}}$ (See Table 1). The retrieval equation is as follows

$$N_{v,\text{trop}} = \frac{N_s - N_{s,\text{strat}}}{M_{\text{trop}}} \qquad (1)$$

More information can be found in the QA4ECV NO$_2$ Product Specification Document (Boersma et al., 2017) and in Zara et al. (2018); Boersma et al. (2018). A preliminary evaluation of the data indicated that QA4ECV NO$_2$ values are 5-20% lower than the earlier version DOMINO v2 of the OMI NO$_2$ data product over polluted regions, and agree slightly better with MAX-DOAS NO$_2$ VCD measurements in Tai'an (China) and De Bilt (The Netherlands) than the DOMINO v2 VCDs (Lorente et al.,

2017; Lorente Delgado, 2019).

The data product files contain a comprehensive amount of metadata. Per pixel the satellite data product provides a total *ex-ante* uncertainty on the retrieved tropospheric VCD, as well as a breakdown of the uncertainty $u_{\text{SAT}}$ into an ex-ante uncertainty budget, with the following uncertainty source components: uncertainty in total SCD $u_{\text{SAT},N_s}$, stratospheric SCD $u_{\text{SAT},N_{s,\text{strat}}}$, and tropospheric AMF $u_{\text{SAT},M_{\text{trop}}}$, which contains contributions from uncertainties in surface albedo $u_{\text{SAT},A_s}$, cloud fraction

(CF) $u_{\text{SAT},f_{cl}}$, cloud pressure $u_{\text{SAT},p_{cl}}$ and a priori profile shape $u_{\text{SAT},S_a}$, and an albedo-CF cross-term (with $c_{A_s,f_{cl}}$ the error correlation coefficient between both properties) (Boersma et al., 2018, section 6).

$$u_{\text{SAT}}^2 = u_{\text{SAT},N_s}^2 + u_{\text{SAT},N_{s,\text{strat}}}^2 + u_{\text{SAT},M_{\text{trop}}}^2$$
$$u_{\text{SAT},M_{\text{trop}}}^2 = u_{\text{SAT},A_s}^2 + u_{\text{SAT},f_{cl}}^2 + u_{\text{SAT},p_{cl}}^2 + u_{\text{SAT},S_a}^2 + 2c_{A_s,f_{cl}}u_{\text{SAT},A_s}u_{\text{SAT},f_{cl}} \qquad (2)$$

Furthermore, the satellite data files provide several relevant instrument parameters, influence quantities (e.g., cloud fraction,

surface albedo, terrain height,...), intermediate quantities (SCD, AMF, stratospheric SCD, ...), and the column averaging kernel $\mathbf{a}_{\text{SAT}}$, which relates the retrieved VCD to the true profile. The a priori NO$_2$ profiles (simulated with TM5) are not stored





in the data files. In the case a user has to adapt a (measured or modelled) profile $\mathbf{x}_h$ at high vertical resolution to the vertical sensitivity of the satellite, he can apply (eq. (11) of Eskes and Boersma, 2003),

$$\mathbf{a}_{\mathrm{SAT}} \cdot \mathbf{x}_h = \mathbf{x}_{h,\mathrm{sm}} \tag{3}$$

where the a priori profile $\mathbf{x}_{\mathrm{SAT},a}$ is not explicit. The dependence of the retrieval on $\mathbf{x}_{\mathrm{SAT},a}$ is already implicit via the averaging

kernel $\mathbf{a}_{\mathrm{SAT}}$.

However, the reference data in the current work are column retrievals or profile retrievals with a limited vertical resolution, and are based on an a priori profile that is different from the satellite retrieval. Before smoothing, satellite and reference retrievals should be adjusted such that they use the same a priori profile (Rodgers and Connor, 2003), therefore knowledge of the satellite a priori profile is relevant. These can be derived from the TM5-MP data files (Huijnen et al., 2010; Williams

et al., 2017), available upon request (see Boersma et al., 2017, for contact details), by spatially interpolating the profiles to the location of the satellite ground pixel.

In this work, we considered data from 2004 up to and including 2016 for the tropospheric VCD and up to and including 2017 for the stratospheric VCD.

### 2.1.2   OMI STREAM stratospheric $NO_2$

The STRatospheric Estimation Algorithm from Mainz (STREAM) (Beirle et al., 2016) was included as an alternative stratospheric estimation scheme in the QA4ECV $NO_2$ data files. In STREAM, the estimate of stratospheric columns is based on satellite observations with negligible tropospheric contribution, i.e. generally over regions with low tropospheric $NO_2$ levels, and for satellite pixels with high clouds, where the tropospheric column is shielded. The stratospheric field is then smoothed and interpolated globally, assuming that the spatial pattern of stratospheric $NO_2$ does not feature strong gradients.

### 2.1.3   NASA OMNO2 data product

Although not the main focus of this work, we do include as benchmark comparisons of an alternative retrieval product, the NASA's OMI $NO_2$ data - OMNO2 version 3.1 (Bucsela et al., 2016; Krotkov et al., 2017) -, with QA4ECV MAX-DOAS. Like QA4ECV OMI $NO_2$, it is also based on the DOAS approach, but nearly all retrieval steps are different between the QA4ECV and NASA OMI $NO_2$ algorithms (Table 1). A detailed comparison between the QA4ECV and NASA fitting approaches

showed small differences between $NO_2$ SCDs (Zara et al., 2018), so differences between the spectral fitting approaches explain only a small part of the differences in the tropospheric VCDs. The stratospheric correction approach differs between the two algorithms. Although the QA4ECV and NASA stratospheric SCDs have not been compared directly, previous evaluations suggest that differences between the approaches typically lead to small but spatially widespread differences of up to $0.5\text{-}1.0 \times 10^{15} \ \mathrm{molec\,cm^{-2}}$ in tropospheric VCDs. This leaves differences between the tropospheric AMF calculations, and especially

the prior information used in their calculations, as the most likely explanation of the lower NASA than QA4ECV $NO_2$ VCDs (e.g., Goldberg et al., 2017).





**Table 1.** OMI satellite data products considered in this work.

| Data product | Spectral fitting | Stratospheric correction | Tropospheric AMF |
|---|---|---|---|
| OMI QA4ECV v1.1 | Zara et al. (2018) | Data assimiliation in TM5-MP (Boersma et al., 2018) | Surface albedo from Kleipool et al. (2008) 5-yr climatology at $0.5° \times 0.5°$; clouds from OMI $O_2$-$O_2$ algorithm (OMCLDO2 data product, Veefkind et al., 2016); a priori $NO_2$ profiles from daily TM5-MP at $1° \times 1°$ |
| OMI STREAM[a] | | Weighted (observations with negligible trop contrib (clean regions, cloudy pixels)) convolution (Beirle et al., 2016) | |
| OMNO2 v3.1 | Marchenko et al. (2015) | Three-step (interpolation, filtering, smoothing) strat field reconstr to fill in the trop contam scenes (Bucsela et al., 2013) | Surface albedo from Kleipool et al. (2008) 5-yr climatology at $0.5° \times 0.5°$; clouds from OMI $O_2$-$O_2$ algorithm (OMCLDO2 data product), a priori profiles from monthly GMI at $1° \times 1.25°$ (Strahan et al., 2013) |

a. OMI STREAM stratospheric VCD is contained in the OMI QA4ECV v1.1 data files.

## 2.2 Ground-based data



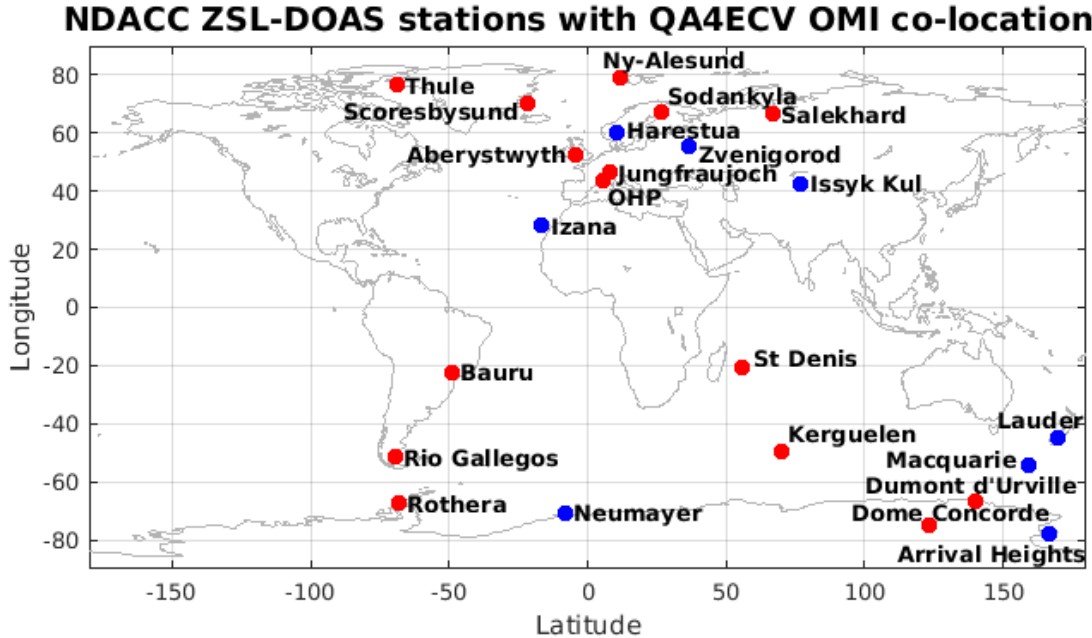

**Figure 1.** Global distribution of the ZSL-DOAS instruments used here. Red markers indicate SAOZ instruments, blue markers other NDACC ZSL-DOAS instruments.

### 2.2.1 Zenith-scattered-light DOAS

The ZSL-DOAS data are part of the Network for the Detection of Atmospheric Composition Change (NDACC) (De Mazière et al., 2018, see also http://www.ndaccdemo.org/), a major contributor to WMO's Global Atmospheric Watch. A significant part of the multi-decadal ZSL-DOAS data is provided by the subnetwork Système d'Analyse par Observation Zénithale (SAOZ;

see Pommereau and Goutail, 1988) from LATMOS, using SAOZ instrumentation in automated data acquisition mode and with fast data delivery.

Zenith-sky measurements are performed during twilight at sunrise and sunset. Due to this measurement geometry with a long optical path in the stratosphere, the measured column is about 14 times more sensitive to stratospheric $NO_2$ than to tropospheric $NO_2$ (Solomon et al., 1987). Moreover, it allows usable measurements during cloudy conditions as well.

Processing followed the NDACC Standard Operation Procedure, as implemented for instance in the LATMOS_v3 SAOZ processing. From slant column intercomparisons, Vandaele et al. (2005) deduce an uncertainty of about 4-7%, but this excludes the uncertainty on the AMF required to convert the slant to vertical columns. Ionov et al. (2008) estimate a total uncertainty on the vertical columns of 21%. A visualisation of the geographical distribution of the instruments is provided in Fig. 1. More details about the particular co-location scheme, taking into account the large horizontal smoothing of these measurements, and

the photochemical adjustment required to convert twilight measurements to satellite overpass times, are provided in Sect. 3.1.





### 2.2.2 Multi axis-DOAS

The tropospheric $NO_2$ VCD data used as a reference are a long-term record of MAX-DOAS (Multi AXis-DOAS) measurements from 10 instruments, reprocessed by different teams for the project QA4ECV (see Table 2). MAX-DOAS measure scattered sunlight under different viewing elevations from the horizon to the zenith (Platt and Stutz, 2008). The observed light

travels a long path (length dependent on the elevation angle) in the lower troposphere, while the stratospheric contribution is removed by a reference zenith measurement. Two different processings of MAX-DOAS data were used for the current validation study, QA4ECV MAX-DOAS and bePRO (Belgian Profiling) MAX-DOAS (Clémer et al., 2010), the latter being part of NDACC.

Thanks to an extensive harmonisation effort within the QA4ECV project, reference QA4ECV MAX-DOAS data sets

were produced by the different teams for all 10 instruments. Those are available at http://uv-vis.aeronomie.be/groundbased/ QA4ECV_MAXDOAS/index.php. This effort was based on a four-step approach (see http://uv-vis.aeronomie.be/groundbased/ QA4ECV_MAXDOAS/QA4ECV_MAXDOAS_readme_website.pdf; Hendrick et al., 2016; Richter et al., 2016; Peters et al., 2017), including (i) the establishment of recommendations for DOAS analysis settings from an intercomparison of $NO_2$ slant column densities retrieved from common spectra; (ii) the development of $NO_2$ AMF look-up tables (LUTs) for harmonising

the conversion of SCDs into VCDs; (iii) the establishment of a first harmonised error budget; (iv) the generation of MAX-DOAS data files in the Generic Earth Observation Metadata Standard (GEOMS) as common format. It is worth noting that since in this QA4ECV approach, only SCDs measured at a relatively high elevation angle (typically $30°$) are used to minimize the impact of aerosols and a priori profile shape on the retrieval, the horizontal location of the centre of the effectively probed air mass is close to the instrument location (typically only  1 km difference).

The second processing, bePRO MAX-DOAS (Clémer et al., 2010; Hendrick et al., 2014; Vlemmix et al., 2015), is available for three BIRA-IASB instruments (at Bujumbura, Uccle and Xianghe). This approach, based on the optimal estimation method (OEM; see Rodgers, 2000), provides profile measurements, albeit with a limited degree of freedom for signal in the vertical dimension, typically ~2 (Bujumbura, Uccle) or ~3 (Xianghe). The horizontal extension of the air masses probed by profile retrieval MAX-DOAS is within about 5 to 15 km distance from the instrument in the viewing direction (Richter et al., 2013a).

The extension depends on the atmospheric visibility (smaller extension for lower visibility) and altitude of the $NO_2$ layer (smaller extension with decreasing profile height). This is in line with typical distances estimated by Irie et al. (e.g.,  2011, Fig. 17). The horizontally projected area of the MAX-DOAS-probed air mass is estimated to be in the order of 0.01 to 0.2 $km^2$ for QA4ECV MAX-DOAS and ~1 $km^2$ for bePRO MAX-DOAS, assuming a $1°$ field-of-view and a simple geometrical approximation.

MAX-DOAS probe the lower troposphere, with the highest sensitivity (described by the column averaging kernel) close to the surface, typically in the lowest 1.5 km of the atmosphere. Nevertheless, the vertical grid extends to ~10 km for QA4ECV MAX-DOAS and ~3 km for bePRO MAX-DOAS.

The MAX-DOAS sites span a wide range of $NO_2$ levels, from relatively low at OHP and Bujumbura (mean tropospheric MAX-DOAS VCD around OMI overpass time ~3 $Pmoleccm^{-2}$) to strongly polluted at Xianghe (mean MAX-DOAS value




**Table 2.** Overview of contributing sources for the QA4ECV MAX-DOAS reference data set.

| Station | Location | Start+end time | Class | Contributor[a] |
|---|---|---|---|---|
| Bremen (DE) | 53.10°N, 8.85°E | 02/2005-12/2016 | Urban | IUP-UB |
| De Bilt (NL)[c] | 52.10°N, 5.18°E | | | |
| Cabauw (NL)[c,d] | 51.97°N, 4.93°E | 03/2011-11/2017 | Sub-urban | KNMI |
| Uccle (BE)[b,d] | 50.80°N, 4.36°E | 04/2011-06/2015 | Urban | BIRA-IASB |
| Mainz (DE)[d] | 49.99°N, 8.23°E | 06/2013-12/2015 | Urban | MPG |
| Observatoire Haute Provence (FR)[d] | 43.94°N, 5.71°E | 02/2005-12/2016 | Rural / Background | BIRA-IASB |
| Thessaloniki (GR)[d] | 40.63°N, 22.96°E | 01/2011-05/2017 | Urban | AUTH |
| Xianghe (CHN)[b,d] | 39.75°N, 116.96°E | 04/2010-01/2017 | Sub-urban | BIRA-IASB |
| Athens (GR)[d] | 38.05°N, 23.86°E | 09/2012-10/2016 | Urban | IUP-UB |
| Nairobi (KEN) | 1.23°S, 36.82°E | 01/2004-11/2014 | Rural / Urban | IUP-UB |
| Bujumbura (BU)[b,d] | 3.38°S, 29.38°E | 01/2014-12/2016 | Sub-urban | BIRA-IASB |

[a] Contributing teams: Aristotle University of Thessaloniki (AUTH), Royal Belgian Institute of Space Aeronomy (BIRA-IASB), Institute of Environmental Physics at University of Bremen (IUP-UB), Max Planck Institute (MPG), Royal Netherlands Meteorological Institute (KNMI). [b] For this sensor also bePRO MAX-DOAS data, providing profile data, is available. [c] The same instrument was operated at two different locations, De Bilt and Cabauw, which are at approximately 30 km distance. [d] An AERONET instrument, measuring aerosol optical depth, is located at this site or at close distance.

~24 Pmolec cm$^{-2}$) (see Fig. 3c, black boxplots), while the other sites are moderately polluted (mean value between 5.6 and 11 Pmolec cm$^{-2}$).

The MAX-DOAS tropospheric VCD is provided with an *ex-ante* uncertainty in the GEOMS data files. Unfortunately the employed uncertainty estimation approach is not harmonised among all data providers. Therefore, we set for QA4ECV MAX-DOAS instead the total uncertainty at 22.2% of the retrieved VCD, following the QA4ECV deliverable D3.9 recommendation (Richter et al., 2016). Following sensitivity tests, aerosol effects (20%) and NO$_2$ a priori profile shape (8%) were identified as the main contributors to the MAX-DOAS uncertainty, while uncorrelated instrument noise is only 2%. However, we do not follow D3.9 (Richter et al., 2016) in its recommended division of the uncertainty into random error and systematic error uncertainty contributions[2] and consider only a total uncertainty. Regarding bePRO MAX-DOAS, we consider 12% total uncertainty for Uccle and Xianghe (following Hendrick et al., 2014), and 21% for Bujumbura (following Gielen et al., 2017). We finally note that for clean sites, an absolute scale uncertainty estimate might be more appropriate.

[2]In D3.9, the systematic error uncertainty is set at 3%, arising from absorption cross-section related systematic error uncertainty on the SCD, while the random error uncertainty is set at 22%, arising from uncertainty on the AMF. However, the assumption that e.g., an error in a priori profile shape would translate to a random error on the retrieved column is not evident in our opinion. In a later analysis (Hendrick et al., 2018), a comparison of QA4ECV MAX-DOAS with more advanced MAX-DOAS profiling methods was performed. This highlighted systematic differences between -12% and +7%, considerably larger than the D3.9-recommended systematic error uncertainty of 3%. This suggests that a larger part of the total uncertainty should be assigned as due to systematic error. Therefore in this work we only consider a total uncertainty of 22.2%, derived from sum in quadrature of the recommended systematic and random components.





As the accuracy of satellite or ground-based remote sensing can be affected by the presence of aerosol, tracking aerosol optical depth (AOD) is useful. The bePRO MAX-DOAS provides aerosol optical depth (AOD) measurements. The QA4ECV MAX-DOAS provides an AOD climatology (Hendrick et al., 2016) based on AERONET (AERONET Aerosol Robotic Network) data (Giles et al., 2019); however, we found that the precision of this climatological data set was inadequate for

the current work, especially for urban sites. Instead, we considered AOD directly from AERONET (Giles et al., 2019) (http://aeronet.gsfc.nasa.gov), whose measurements are based on Cimel Electronique Sun–sky radiometers. Level 2.0 AOD at wavelength 440 nm was chosen, which is within the QA4ECV MAX-DOAS retrieval window of 425 - 490 nm. Note that the AERONET data is already cloud filtered.

## 3  Validation

### 3.1  Validation methodology

The generic validation protocol is similar to that outlined by Keppens et al. (2015), and tailored within the QA4ECV project for the ECVs $NO_2$, HCHO and CO (Compernolle et al., 2018). Terms and definitions applicable to the quality assurance of ECV data products have been agreed upon within QA4ECV (Compernolle et al., 2018); the full set can be found at Compernolle and Lambert (2017). The discussion and analysis on comparison error follows the terminology and framework detailed within

the GAIA-CLIM project (Verhoelst et al., 2015; Verhoelst and Lambert, 2016).

In the following sections, we detail the baseline validation methodology.

### 3.1.1  Screening criteria

Filters to the satellite data product are applied following the recommendations in the QA4ECV $NO_2$ product specification document (PSD) (Boersma et al., 2017), and to minimize comparison error with MAX-DOAS.

Following the QA4ECV $NO_2$ product specification document (PSD) (Boersma et al., 2017), satellite data is kept for tropospheric $NO_2$ validation if the following conditions are met:

– (1) no raised errorflag,

– (2) satellite solar zenith angle (SZA) $< 80°$,

– (3) the so-called 'snow-ice flag' indicating either 'snow free land', or 'ice free ocean' or a sea-ice coverage below 10%

– (4) the ratio of tropospheric AMF over geometric AMF, $\frac{M_{\mathrm{trop}}}{M_{\mathrm{geo}}}$, must be higher than 0.2, to avoid scenes with very low tropospheric AMF (typically occurring when the TM5 model predicts a large amount of $NO_2$ close to the surface in combination with aerosols or clouds effectively screening this $NO_2$ from detection)  and

– (5) effective cloud fraction (CF) < 0.2. This last filter is comparable to the PSD recommendation of cloud radiance fraction (CRF) < 0.5, and was chosen because effective cloud fraction is a more general property than CRF. Note that the





effective cloud properties cloud fraction and cloud height are sensitive to both aerosol and cloud (Boersma et al., 2004). It should be mentioned that cloudy pixel retrievals - although subject to larger errors compared to clear-sky pixels - can still be used (e.g., in data assimilation), provided the averaging kernel is taken into account (Schaub et al., 2006).

- (6) Not mentioned in the PSD, but applied by Boersma et al. (2018), is a filter to limit the impact of aerosol haze and low clouds. In the latter work, this was accomplished by excluding ground pixels with a high retrieved cloud pressure, i.e. $p_c > 850$ hPa. Unfortunately, this filter can remove a substantial portion of the data, therefore a less strict filter was searched for in the current work. A low cloud can lead to a high uncertainty in the retrieved tropospheric $NO_2$ value when it is uncertain if it is located above the trace gas (mainly a screening effect and therefore a low AMF) or is at similar height (partial screening effect, partial surface albedo effect, and therefore a higher AMF). This is registered in the uncertainty component due to cloud pressure $u_{SAT,\,p_c}$ available within the data product. Data analysis reveals that for several sites (Xianghe, Uccle, De Bilt, Bremen, Athens), a relatively small number of ground pixels are responsible for an important contribution to the root-mean-square (RMS) of the ex-ante satellite uncertainty, via the cloud pressure component $u_{SAT,\,p_c}$. Most of these high-uncertainty ground pixels have a low retrieved effective cloud pressure (Fig. S1 in supplement), indicative of aerosol haze or low-lying cloud. The aforementioned cloud pressure filter used by Boersma et al. (2018) would effectively remove these suspicious ground pixels, but also many other pixels with a low $u_{SAT,\,p_c}$. Therefore, we chose instead to apply as filter (6) a one-sided sigma-clipping on $u_{SAT,\,p_c}$: data where $u_{SAT,\,p_c,i} > \mathrm{mean}(u_{SAT,\,p_c,i}) + 3 \times \mathrm{SD}(u_{SAT,\,p_c,i})$ are removed. This sigma-clipping removes a smaller percentage of the data, while still achieving its goal of limiting $u_{SAT,\,p_c}$ and $u_{SAT}$. After this filtering step, $u_{SAT,\,p_c}$ is only a minor contributor to the OMI uncertainty budget.

- (7) Finally, satellite ground pixels with a footprint $> 950\,\mathrm{km}^2$, corresponding to the 5 outermost rows at each swath edge of the OMI orbit, are removed to limit the horizontal smoothing difference error with the MAXDOAS data. Filter (7) is not a filter on satellite data quality, but rather a limit on the scope of the validation.

Regarding stratospheric $NO_2$ validation, only filters (1)-(3) are applied. Hence both cloudy and non-cloudy scenes are used. Regarding the OMNO2 data product, we followed the recommendation of Bucsela et al. (2016) by only including ground pixels for which the least-significant bit of the variable VcdQualityFlags is zero (indicating good data). Furthermore, the effective cloud fraction must be $< 0.2$ and the pixel area $< 950\,\mathrm{km}^2$.

No screening was applied to the ground-based reference data sets. In particular, filtering on the MAX-DOAS cloud flag is not applied as baseline as it is not available for all data sets. It should be noted that clouds can impact the quality of MAX-DOAS retrievals (see e.g., radiative transfer simulations of Ma et al., 2013; Jin et al., 2016).

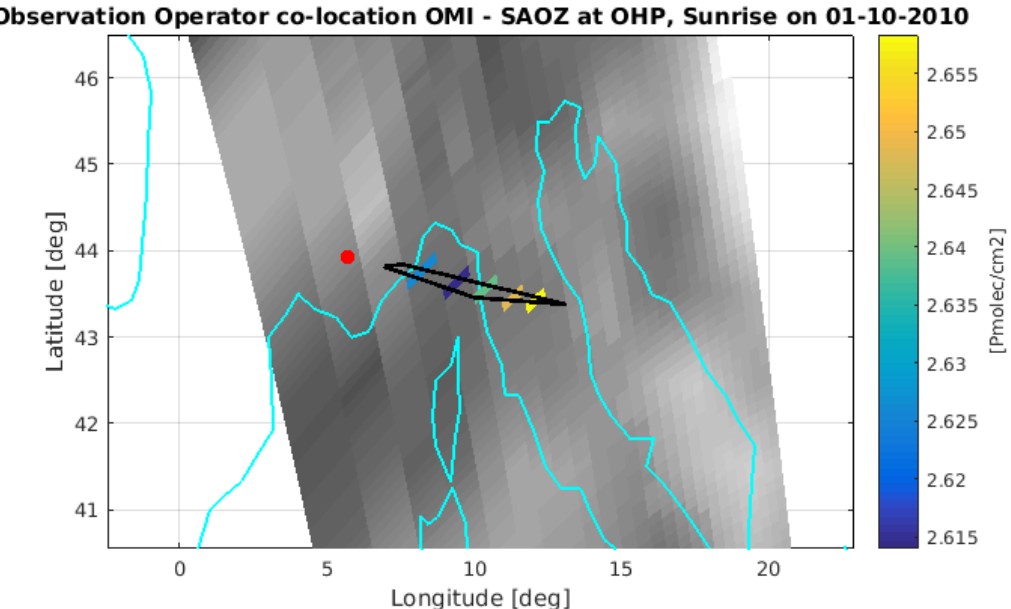

**Figure 2.** Illustration of a single co-location between OMI and a sunrise ZSL-DOAS measurement using the dedicated observation operator. The red dot marks the location of the ground instrument, the cyan lines the coast lines of this part of the Mediterranean. The grey-scale background contains the full orbit data, the coloured pixels are those that have their center within the observation operator (black polygon), i.e. those that are averaged to obtain a satellite measurement comparable to that of the ZSL-DOAS instrument.

### 3.1.2 Co-location criteria and processing

**Stratospheric column**

The airmass to which a ZSL-DOAS measurement is sensitive spans over many hundreds of kilometers towards the rising or setting Sun (e.g. Solomon et al., 1987). The co-location scheme employed here takes this into account by averaging all OMI ground pixels of a temporally co-located orbit (maximum allowed time difference of 12 hours) which have their center within the ZSL-DOAS observation operator. This 2-D polygon is a parametrization of the actual extent of the airmass to which the ZSL-DOAS measurement is sensitive. Its horizontal dimensions were derived using a ray tracing code, mapping the 90% interpercentile of the stratospheric vertical column to a projection on the ground and then parametrized as a function of the solar zenith and azimuth angles during the twilight measurement, where the SZA during a nominal single measurement sequence is assumed to range from 87° to 91° (at the location of the station). Note that the station location is not part of the area of actual measurement sensitivity. The average OMI stratospheric column over this observation operator can then be compared to the column measured by the ZSL-DOAS instrument. An illustration of a single such co-location is presented in Fig. 2. Note that at polar sites, the above mentioned SZA range may not be covered entirely. For more details, we refer to Lambert et al. (1996) and Verhoelst et al. (2015).



To account for effects of the photochemical diurnal cycle of stratospheric $NO_2$, the ZSL-DOAS measurements, obtained twice daily at twilight at each station, are adjusted to the OMI overpass time using a model-based factor. The latter is extracted from LUTs calculated with the PSCBOX 1D stacked-box photochemical model (Errera and Fonteyn, 2001; Hendrick et al., 2004) initiated by daily atmospheric composition and meteorological fields fields from the SLIMCAT chemistry-transport

5   model (Chipperfield, 1999). The amplitude of the adjustment depends strongly on the effective SZA assigned to the ZSL-DOAS measurements. It is taken here to be 89.5°. The uncertainty related to this adjustment is of the order of 10% or 1 to 2 $10^{14}\,\mathrm{molec\,cm}^{-2}$.

**Tropospheric column**

Regarding the tropospheric column validation, satellite data is kept if the satellite ground pixel covers the MAX-DOAS

10   instrument location, and if a MAX-DOAS measurement is within a 1-hour interval centered at the satellite measurement time. The average of all MAX-DOAS measurements within this 1-hour interval is taken. The typical number of MAX-DOAS measurements taken within this time interval was 2-4 for most sites. This procedure was applied to both QA4ECV OMI $NO_2$ and the OMNO2 comparisons.





## 3.2 Impact of quality screening

Quality screening is a necessary step before a satellite data product can be used, but it can be a limit to the data product's scope. Fig. 3a presents the remaining fractions of satellite overpass data at the MAX-DOAS sites at each of the 7 successive filter steps described in section 3.1.1. Note that the sites Cabauw and De Bilt are not included, as the results are very close to that of

Uccle.

The error flag (1) removes ~10-30% of the data, filters on SZA and snow-ice flag (2, 3) have a relatively small impact, the filter on AMF ratio (4) has a large impact on the sites Bremen, Mainz, Cabauw, De Bilt, Uccle and Xianghe (35-40% of data removed), and finally the filter on CF (5) has an important screening impact on all sites (see Fig. 3), removing up to 60% of the data at the site of Bujumbura. As an alternative to the CF filter, we tested also the CRF<0.5; for most sites the CRF and

CF filter have a near identical impact, but for Bujumbura and Nairobi the CRF filter is more restrictive (results not shown). In combination, the PSD-recommended quality filters (filters (1) to (5)) remove between 56% (Athens) and 90% (Bremen) of the data.

Filter (6), the filter on the uncertainty component due to cloud pressure $u_{SAT, p_c}$, removes at most 5% of data, at the site of Xianghe, while the alternative filter on cloud pressure would have removed 15% of data (Fig. S1). The filter on ground pixel

size (7) removes 3-16% additional data.

The screening can have a strong seasonal effect; for example, the winter months are strongly underrepresented for the West-European urban sites (Fig. 3b). Fig. 3c presents, per MAX-DOAS site, box plots of co-located MAX-DOAS tropospheric $NO_2$ measurements before (black) and after (blue) screening. Both mean and median value decrease by the filtering step. We conclude that the quality screening tends to reject scenes with a high tropospheric $NO_2$ VCD, i.e., the restriction to

quality-screened scenes leads to a negative sampling bias with respect to the ensemble of all scenes. In absolute scale, the screening effect is the strongest at the site Xianghe, leading to a reduction in yearly mean tropospheric $NO_2$ from 24 to 15 $Pmolec\,cm^{-2}$ (40% decrease). At Nairobi, Thessaloniki, Bremen, De Bilt and Cabauw, the tropospheric VCD is reduced by several $Pmolec\,cm^{-2}$. The cloud filter is a main contributor to this sampling bias. This is in accordance with the results of Ma et al. (2013), where higher tropospheric $NO_2$ was measured by MAX-DOAS in Beijing in cloudy conditions compared

to clear-sky conditions. Indeed, cloudy conditions lead to less photochemical loss of tropospheric $NO_2$, as explained by with model results (Boersma et al., 2016). In comparisons of OMI tropospheric $NO_2$ with independent data, care should be taken that the independent data is also sampled for clear-sky conditions (Boersma et al., 2016). A systematic influence of clouds on the MAX-DOAS retrievals might contribute to the observed sampling bias effect.

It can be argued that the AMF ratio filter (filter (4)) is too restrictive. In section S2 results are presented for the less restrictive

$\frac{AMF_{trop}}{AMF_{geo}} \geq 0.05$. The remaining data fraction is slightly increased at the sites Bremen, Mainz, Uccle, De Bilt and Cabauw (from ~8% to ~10%) and the winter months are better represented (see Fig. S2). The negative sampling bias at De Bilt and Bremen is reduced and removed at Mainz. As will be shown in section 3.4.6, this adapted filtering generally has no negative impact on the satellite vs MAX-DOAS comparisons.



**Figure 3.** a) Starting from satellite data with ground pixel covering the MAXDOAS site, the remaining data fraction after applying each of the 7 filter criteria is presented. The criteria are explained in section 3.1.1. The sites Cabauw and De Bilt are not included here, as the fractions are very close to that of Uccle. b) Remaining fraction per month, after applying all filters. c) Per site, boxplots of QA4ECV MAX-DOAS data co-located with QA4ECV OMI, before applying the filters (black), after applying the filters (blue), and of QA4ECV OMI co-located with MAX-DOAS and after applying the filters (red). The sites are sorted according to the median MAX-DOAS value before filtering. Boxplot legend: box edges: 1st and 3th quartiles; orange line: median; green cross: mean; whiskers: 5th and 95th percentiles.

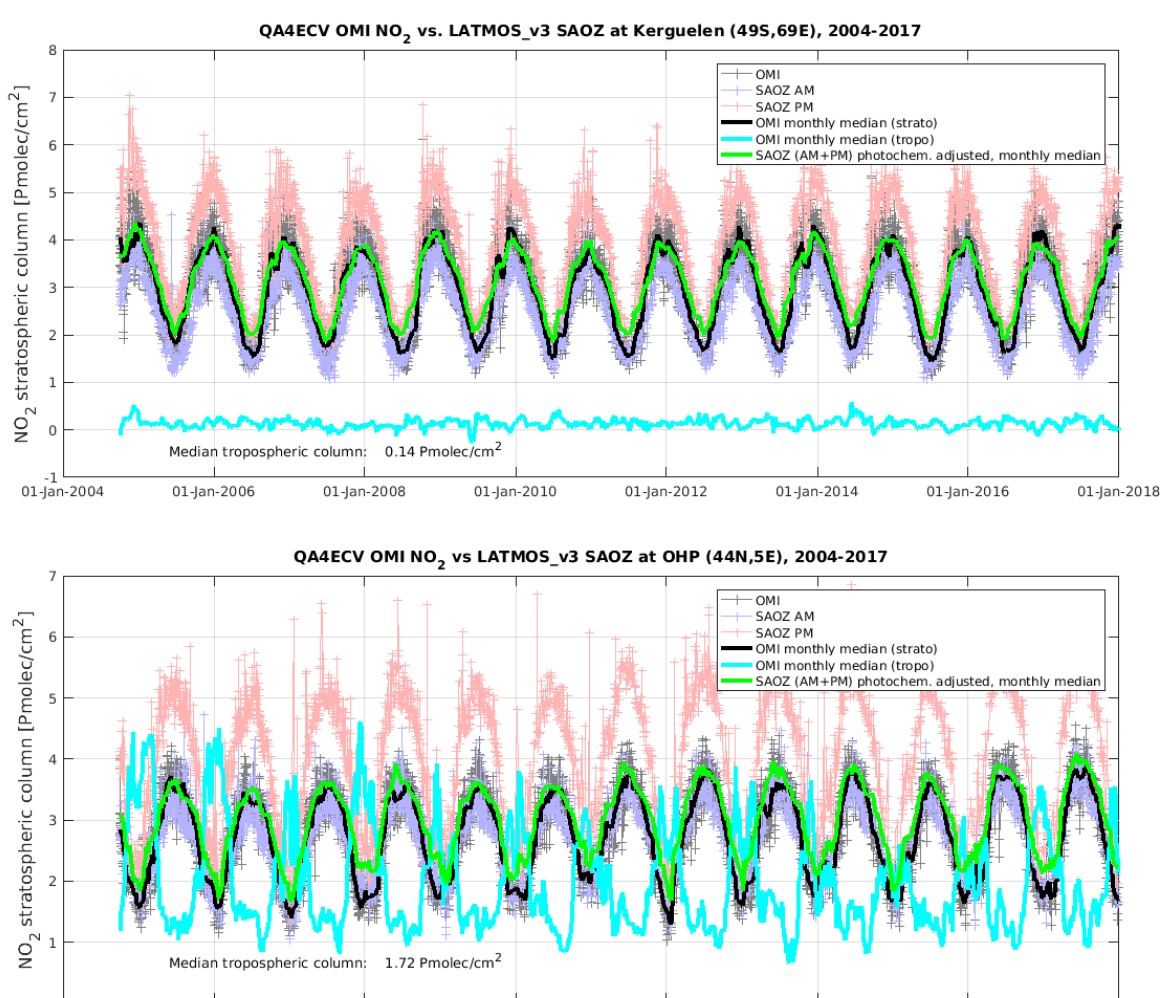

**Figure 4.** *Upper panel:* Time series of OMI and SAOZ stratospheric NO$_2$ above the NDACC station of Kerguelen in the Indian Ocean, typical for clean background conditions. *Lower panel:* Similar to the upper panel but for the Observatoire de Haute Provence, which shows more significant tropospheric columns in winter due to pollution.

## 3.3 Comparison of OMI stratospheric NO$_2$ with ZSL-DOAS

Figure 4 contains time series of stratospheric NO$_2$ columns, from both satellite (QA4ECV product) and ground-based instruments, at two illustrative ground sites: Kerguelen in the Southern Indian Ocean, which is representative for very clean background conditions, and the Observatoire de Haute Provence in France, which is affected by significant tropospheric pollution in local winter, often exceeding the wintertime stratospheric column. The graphs show the well-known seasonal cycle in stratospheric NO$_2$, which is captured similarly by satellite and ZSL-DOAS instrument. Already evident from perusal of the





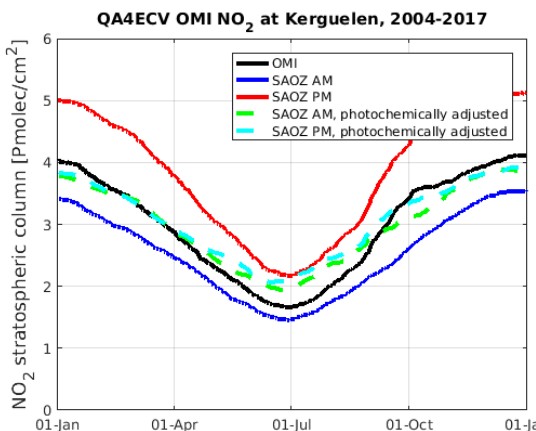 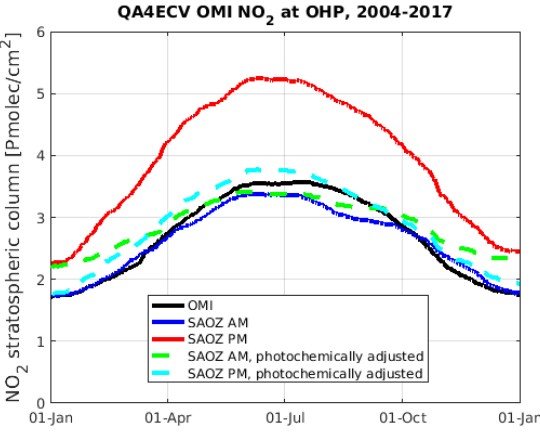

**Figure 5.** Climatological, i.e., all years mapped to a single year and with a 1-month smoothing function applied, comparison between QA4ECV OMI stratospheric $NO_2$ and the ZSL-DAOS instruments at Kerguelen and OHP, revealing overall good agreement.

results at OHP, is that the stratospheric comparison is hardly affected by the peaks in tropospheric pollution, e.g. in winter 2005-2006, indicating a good separation between troposphere and stratosphere in the QA4ECV OMI retrievals.

To better reveal differences in representation of the seasonal cycle, Fig. 5 contains a mapping of the full time series at these two stations to a single "average" year, with a 1-month smoothing function applied. While the seasonal cycle is in general well represented, with accurate levels in local summer, the QA4ECV OMI stratospheric $NO_2$ column does appear to be systematically lower than the ground-based value in local winter, at these two sites. The QA4ECV $NO_2$ retrieval includes a 2nd order a posteriori temperature correction to adjust for the difference in absorption cross section between the assumed 220K and the true effective temperature (Zara et al., 2017). The ZSL-DOAS data however were not temperature corrected and Hendrick et al. (2012) estimate the impact to range between a 2.4% overestimation in local winter and a 3.6% underestimation in local summer for ZSL-DOAS measurements at Jungfraujoch. In other words, the amplitude of the seasonal cycle should be about 6% larger than now reported by the ZSL-DOAS. This could explain a significant part of the discrepancy between satellite and ground seasonal cycle at these two sites, but requires confirmation with a proper ZSL-DOAS temperature correction. Development work on this is ongoing (Hendrick, priv. comm.) but beyond the scope of the current paper. The excellent agreement between sunrise and sunset ZSL-DOAS measurements after mapping to the OMI overpass time at Kerguelen suggests the photochemical adjustment to work well, but it does not exclude the presence of biases that are common to sunrise and sunset measurements.

Fig. 6 presents the network-wide results in terms of bias and comparison spread per station as a function of latitude. On average, QA4ECV OMI stratospheric $NO_2$ seems to have a minor negative bias (-0.2 Pmolec/cm$^2$) w.r.t. the ground-based network, related mostly to differences in local winter as described above. In view of the station-to-station scatter of the order of 0.3 $Pmolec/cm^2$ and the uncertainties on the ground-based data, this is hardly significant and it is roughly in line with validation results for other data sets of OMI stratospheric $NO_2$ (e.g. Celarier et al., 2008; Dirksen et al., 2011). Interestingly, the





STREAM stratospheric $NO_2$ product, also included in the data files but based on a very different approach (Beirle et al., 2016), does not present this negative bias (see lower panel in Fig. 6). This deserves further exploration but that is outside the scope of the current paper. The comparison spread at a single station varies from 0.2 Pmolec/cm$^2$ to 0.5 Pmolec/cm$^2$, corresponding to about 10% of the stratospheric column. Raw comparisons at Zvenigorod, Russia, yielded a higher comparison spread (1.2

5    Pmolec/cm$^2$) due to very large pollution events in the Moscow area affecting the ZSL-DOAS measurements, but for Fig. 6 these were excluded by filtering out co-located pairs with an OMI tropospheric column larger than 3 Pmolec/cm$^2$.

Returning to the issue of a potential stratospheric temperature (and hence seasonal) dependence in the differences, Fig. 7 contains a pole-to-pole graph of the bias between QA4ECV OMI stratospheric $NO_2$ and sunset ZSL-DOAS, separated by season. No clear seasonal signature can be observed here across all latitudes, at least not one that is significant w.r.t the station-

10    to-station scatter or w.r.t. the standard deviation of the difference within a season (the error bars on the graph).



**Figure 6.** Meridian dependence of the mean (the circular markers) and standard deviation ($\pm 1\sigma$ error bars) of the individual differences between QA4ECV (upper panel) and STREAM (lower panel) OMI stratospheric $NO_2$ column data and ZSL-DOAS reference data, represented at individual stations from the Antarctic to the Arctic. The values in the legend correspond to the mean and standard error of all mean (per station) differences.

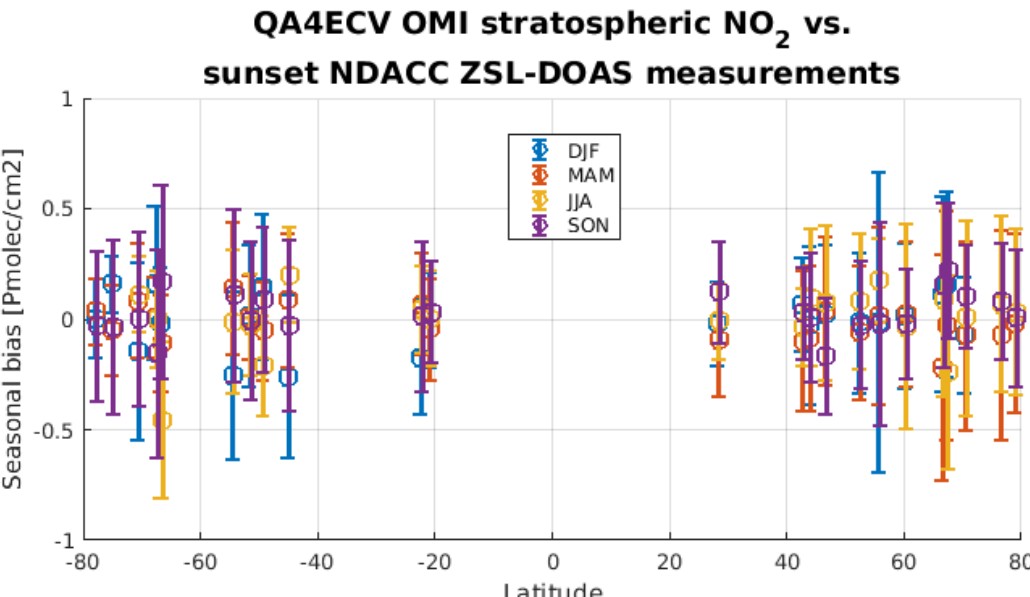

**Figure 7.** Meridian dependence of the mean and standard deviation (error bars) of the differences between QA4ECV OMI stratospheric $NO_2$ column data and sunset ZSL-DOAS reference data, bias-corrected for the annual mean difference, represented at individual stations from the Antarctic to the Arctic and separated by season.





### 3.4 Comparison of OMI tropospheric $NO_2$ with MAX-DOAS

A key issue in the geophysical validation of satellite data sets with respect to sub-orbital reference measurements are the additional uncertainties that appear when comparing different perceptions of the inhomogeneous and variable atmosphere, that is, when comparing data sets characterized with different sampling and smoothing properties, both in space and time, a
main topic of the European project GAIA-CLIM (Verhoelst et al., 2015; Verhoelst and Lambert, 2016). Potential comparison error sources for satellite vs MAX-DOAS are discussed in sections 3.4.1 to 3.4.5, following the framework and terminology of Verhoelst et al. (2015); Verhoelst and Lambert (2016). The impact of horizontal smoothing difference error on the bias is presented in a qualitative way in Figs. 8 and S5-S8.

Comparison results of QA4ECV OMI with MAX-DOAS are provided in section 3.4.6. Overall bias and dispersion are
provided in boxplots of the differences per site (Fig. 9); here also comparisons of the NASA OMI data product OMNO2 with MAX-DOAS are shown. The seasonality of the bias for each site is shown in Figs. 10 and 11. Fig. 12 presents the overall discrepancy between QA4ECV OMI and MAX-DOAS as given by the mean-squared deviation (MSD), split into bias, seasonally cyclic and residual components. This figure also presents the consistency of the RMSD with the combined ex-ante uncertainty. The impact of adapting screening criteria on bias and dispersion is shown in Fig. S9-S13. A priori profile
harmonization and vertical smoothing is presented in Fig. 13 for the bePRO sites Uccle and Xianghe. The discussion of these figures is point-by-point given in section 3.4.6. Table 3 gives an overview of the error source attributions.

### 3.4.1 Comparison error sources: overview

Part of the discrepancies between the OMI and the MAX-DOAS data sets are due to comparison errors. Starting from the general comparison equation (Verhoelst et al., 2015; Verhoelst and Lambert, 2016), the difference between satellite and reference
measured values can be approximated in this specific case as

$$N_{v,\text{trop,SAT}} - N_{v,\text{trop,REF}} = e_{\text{total}} = -e_{\text{REF}} + e_{\text{SAT}} + e_{Sr} + e_{\Delta r} + e_{\Delta t} + e_{\Delta z} \tag{4}$$

with $N_{v,\text{trop,SAT}}$ and $N_{v,\text{trop,REF}}$ tropospheric VCD values measured by satellite and reference ground-based sensors respectively, $e_{\text{SAT}}, e_{\text{REF}}$ the errors in both measurements, $e_{Sr}$ the horizontal smoothing difference error (as the horizontal projection of the probed air mass of satellite and ground-based measurement is different), and $e_{\Delta r}, e_{\Delta t}, e_{\Delta z}$ the horizontal, temporal and vertical
sampling difference error (as satellite and ground-based measurement are not taken at exactly the same space and time).

### 3.4.2 Temporal sampling difference error

The temporal sampling difference error, and MAX-DOAS uncorrelated random error, are already mitigated by averaging the MAX-DOAS measurements within a 1.0 h interval. We found that using larger time intervals can lead to an increase in the bias, likely because of photochemical evolution and transport of the $NO_2$ molecule, but at this small time window the temporal
sampling difference error has a random character[3]. The residual uncertainty can be estimated by taking the uncertainty of the

---

[3]This is checked by comparing MAX-DOAS measurements before and after the satellite overpass time, for the different overpasses.





mean of the MAX-DOAS values within each time interval. Subtracting in quadrature this component from the RMSD, the $N_{v,\text{trop,SAT}}$-$N_{v,\text{trop,REF}}$ discrepancies at the different sites would be reduced by less than $0.1\ \text{Pmolec cm}^{-2}$ for the sites OHP, Bujumbura, Athens and Nairobi, and by 0.1 to at most $0.5\ \text{Pmolec cm}^{-2}$ for the other sites. Temporal sampling difference error and MAX-DOAS uncorrelated random error can therefore be considered as insignificant contributions to the $N_{v,\text{trop,SAT}}$-

$N_{v,\text{trop,REF}}$ discrepancies, and are not discussed further here. In agreement with this, Wang et al. (2017) found that the impact of temporal sampling difference error on satellite vs. MAX-DOAS tropospheric $NO_2$ VCD comparisons was negligible.

### 3.4.3   Horizontal sampling difference error

Tropospheric $NO_2$ has a large spatial variability, especially at polluted sites, therefore random and systematic features in the true $NO_2$ field at the scale of the distance between MAX-DOAS location and co-located satellite ground pixel (typically a few

to a few tens of km, ~10-14 km on average) can be expected. However, one must realize that (i) there is no directional preference in the co-locations, therefore directional features are averaged out in the comparison, and (ii) the satellite measurements are strongly spatially smoothed.

To estimate the impact of horizontal sampling difference error, we compare two sets of QA4ECV OMI $NO_2$ tropospheric VCDs. Regarding the first set ($N_{v,\text{trop,SAT1}}$), it is required that its ground pixel covers the MAX-DOAS site and its pixel center is

within 5 km from the site. The second set ($N_{v,\text{trop,SAT2}}$) has its ground pixel second-nearest to the site, within the same overpass. $\text{SAT}_1$ pixels are on average at 3-4 km from the site and $\text{SAT}_2$ pixels at 11-12 km, while the distance between $\text{SAT}_1$ and $\text{SAT}_2$ pixels is typically 13.6 km, i.e., comparable to the mean distances encountered in the OMI vs. MAX-DOAS comparisons. Note that the discrepancy between $N_{v,\text{trop,SAT1}}$ and $N_{v,\text{trop,SAT2}}$ is due to both horizontal sampling difference error and to random noise error.

Details of the analysis are in section S3 of the supplemental material. The main conclusions are as follows.

- The bias caused by horizontal sampling difference error reaches at most ~-0.6 $\text{Pmolec cm}^{-2}$ (at Athens, Bremen and Mainz), and is therefore only a very minor contributor to the observed bias between OMI and MAX-DOAS (discussed later in section 3.4.6).

- The dispersion of $N_{v,\text{trop,SAT2}} - N_{v,\text{trop,SAT1}}$ can in principle be due to variation in total slant column, in AMF or in

stratospheric slant column (see Eq. (1)). It is shown in the supplement that it is largely due to variation of the slant column. It follows that uncorrelated random noise error mainly originates from the slant column, not from AMF or stratospheric column (since these do not vary much between neighbouring pixels). This then justifies the use of the ex-ante uncertainty component due to SCD uncertainty, $u_{\text{SAT},N_s}$, as an estimate of the total random error uncertainty. Note that $u_{\text{SAT},N_s}$ was scaled such that it only accounts for random error of the slant column (Zara et al., 2018), not for

systematic error.

- At the sites Bujumbura and Nairobi, $u_{\text{SAT}_1,N_s}^2 + u_{\text{SAT}_2,N_s}^2$ exceeds the variance of the difference, indicating that $u_{\text{SAT},N_s}$ is sometimes overestimated.





- The standard deviation caused by horizontal sampling difference error (obtained by subtracting in quadrature the dispersion due to random noise) is minor compared to the discrepancies encountered in the OMI vs. MAX-DOAS comparisons.

### 3.4.4 Vertical sampling difference error

Two sources of vertical sampling difference error can be identified. First, the surface altitude of the ground-based MAX-DOAS sensor, and the mean surface altitude of the OMI ground pixel, are not exactly the same. To estimate a correction, we applied a VMR-conserving vertical shift of the satellite a priori profile, described by Zhou et al. (2009). This hardly changed $N_{v,\text{trop,SAT}} - N_{v,\text{trop,REF}}$ (bias changes of 0.3 $\mathrm{Pmolec\,cm}^{-2}$ or less). This VMR-conserving approach probably underestimates the discrepancy at the sites Athens and Bujumbura which have a complicated orography. The MAX-DOAS instrument at Athens is located on one of the hills surrounding the city at 527 m altitude, while the mean surface altitude of the co-located satellite pixels is ~200m. The MAX-DOAS measurement therefore misses the lowest part of the tropospheric column; correcting for this would increase the already negative bias. The MAX-DOAS instrument at Bujumbura is at 860 m altitude, at the edge of the city which is located in a valley surrounded by 2000-3000m high mountains (Gielen et al., 2017); this causes the mean surface altitude of the co-located satellite pixels (~1.2 km) to be higher than the MAX-DOAS instrument.

A second source of vertical sampling difference error is the fact that the MAX-DOAS only measures the lower tropospheric $NO_2$ VCD, while the satellite measures the full tropospheric VCD. This is, in principle, a source of positive bias in $N_{v,\text{trop,SAT}} - N_{v,\text{trop,REF}}$ and therefore cannot explain the observed negative bias in the comparison. A proper quantification of this bias source depends critically on the assumed vertical profile shape and is out of scope of the current work.

### 3.4.5 Horizontal smoothing difference error

Ideally, subpixel variation of the tropospheric VCD would be estimated using a high resolution model with grid cell area comparable to the MAX-DOAS horizontally projected area of the probed air mass. Instead, we employ here two semi-quantitative approaches to estimate the bias from horizontal smoothing difference error.

In the first approach, the horizontal smoothing effect is estimated from the QA4ECV OMI $NO_2$ data itself. 'Superpixel' OMI tropospheric VCDs are constructed by averaging OMI VCDs of individual pixels of relatively small size ($\leq 500\,\mathrm{km}^2$) within a 20 km radius centered at the MAX-DOAS site. Per overpass, such a superpixel VCD is compared with the individual ground pixel VCD covering the MAX-DOAS site. With this procedure, a superpixel consists on average of 3 individual ground pixels. The mean difference per season, over the years 2004-2016, is presented in Fig. 8. The second approach is similar, but uses S5p TROPOMI $NO_2$ data (May 2018 to May 2019, RPRO (reprocessed) + OFFL (offline) data with processor version 01.02.00-01.02.02), and the superpixel tropospheric VCD is constructed by averaging, per overpass, VCDs of individual pixels that are within a latitude, longitude box of $\Delta\text{lat} = 0.14°$, $\Delta\text{lon} = 0.7°$ centered at the MAX-DOAS site. TROPOMI has a similar overpass time as OMI (early afternoon) and a considerably finer resolution ($3.5 \times 7\text{km}^2$ at nadir). The area of this superpixel corresponds to ~700-900 $\text{km}^2$, i.e., about the size of a bigger OMI pixel, and contains typically 20 TROPOMI ground pixels.



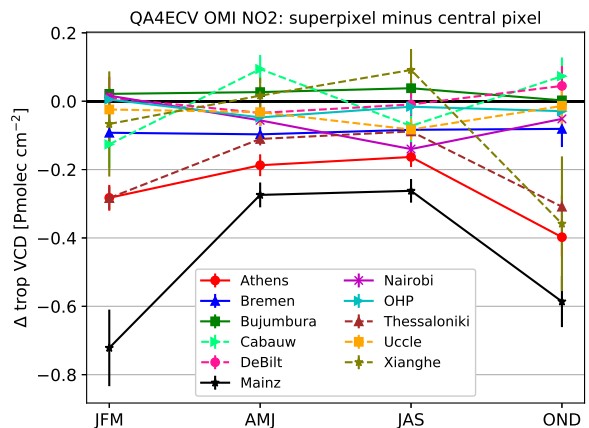
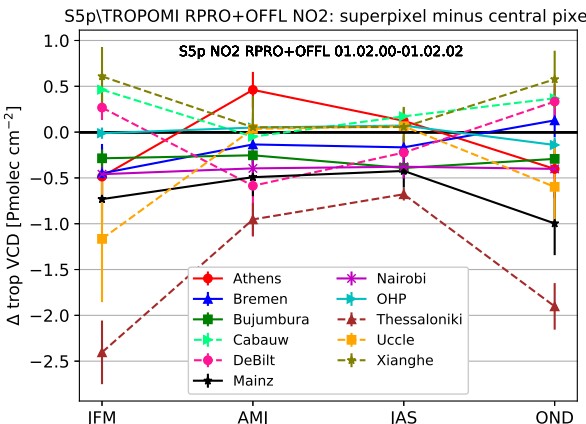

**Figure 8.** a: Mean difference, per season, between QA4ECV OMI superpixel (ground pixels averaged within 20 km of the central site) and the central OMI ground pixel, using data in the time range 2004-2016. b: Similar as (a), but using TROPOMI $NO_2$ data (time range 04/2018-05/2019, and the superpixel is defined within a latitude, longitude box of $\Delta\mathrm{lat} = 0.14°$, $\Delta\mathrm{lon} = 0.7°$ centered at the MAX-DOAS site.

The OMI-based approach has as advantage that the time range is appropriate but it is limited by the large ground pixel size. Regarding the finer-resolution TROPOMI data, one should keep in mind that its ground pixel size is still large compared to the horizontally projected area of probed air mass of the MAX-DOAS[4], hence the contribution of horizontal smoothing difference error to bias and comparison might still be underestimated. Another limitation is that the considered TROPOMI time range does not overlap with the considered time range of OMI. Both approaches suggest a negative bias contribution due to horizontal smoothing difference error at the sites Mainz and Thessaloniki and no such bias contribution at OHP, while for other sites the results are mixed (bias is varying over the seasons, and/or different results between the OMI and TROPOMI-based calculations).

A tropospheric $NO_2$ monthly field with sub-pixel variability is derived from the QA4ECV OMI $NO_2$ data, using a variant of the temporal averaging approach of Wenig et al. (2008)[5], and visualised in Fig. S5-S8, for months with a minimal (left column) and maximal (right column) OMI vs. MAX-DOAS bias (as derived from Figs. 10-Figs. 11). Fields are constructed for each month, by averaging over the years 2004-2016. The resulting field is horizontally smoothed; the variability is an underestimate of the true horizontal $NO_2$ variability. Sub-pixel enhanced tropospheric $NO_2$ approximately centered at the MAX-DOAS site can be identified in high-bias months at Nairobi, Thessaloniki and Mainz, while for the sites OHP, Bujumbura, Uccle, De

---

[4]The horizontal distance of the QA4ECV MAX-DOAS measurements is small compared to a TROPOMI pixel in both the viewing and the perpendicular direction. Regarding bePRO MAX-DOAS, while having a small field-of-view, its probed distance in the viewing direction (~10 km) is of similar or slightly larger magnitude as the cross-section of a TROPOMI ground-pixel.

[5]Here, Per $0.02 \times 0.02$ grid cell the arithmetic average of covering satellite ground pixels is taken, rather than a weighted average as done by Wenig et al. (2008). Only ground pixels with area $< 950\mathrm{km}^2$ are considered.





Bilt/Cabauw and Xianghe this is clearly not the case. At Athens the pollution peak centre is at some 10 km from the sensor, and for Bremen no clear peak is identified.

The contribution of horizontal smoothing difference error to the (OMI - MAX-DOAS) bias at Mainz is consistent with the results of Drosoglou et al. (2017), who achieved a significant bias reduction by adjusting the OMI data with factors derived
from air quality simulations at a high spatial resolution of 2 km.

Similar maps were constructed by Ma et al. (e.g., 2013); Chen et al. (e.g., 2009) to estimate the impact of the horizontal smoothing effect on satellite vs. DOAS comparisons.

### 3.4.6  Comparison results

**Bias and dispersion.** Fig. 9 (black boxplots) presents, per MAX-DOAS site, boxplots of the difference of QA4ECV OMI
with co-located QA4ECV MAX-DOAS. At all sites, the bias of QA4ECV OMI with respect to QA4ECV MAX-DOAS is negative. In absolute scale, it is the smallest at the lower-pollution sites OHP and Bujumbura (mean difference -0.9 and -1.7 $\mathrm{Pmolec\,cm^{-2}}$ respectively), and largest at the sites Thessaloniki and Mainz (mean difference of ~-4 $\mathrm{Pmolec\,cm^{-2}}$). In relative scale, the bias is smallest (median relative difference between -15 to -20%) at the sites Uccle, Cabauw, De Bilt and Xianghe and largest (median relative difference ~-70%) at Bujumbura and Nairobi. The difference dispersion, expressed as interquartile
range (IQR) is smallest at the sites Bujumbura, OHP and Nairobi (~1-2 $\mathrm{Pmolec\,cm^{-2}}$) and largest at the sites Mainz and Xianghe (~5-6 $\mathrm{Pmolec\,cm^{-2}}$). As discussed in sections 3.4.2 to 3.4.5, among the different comparison error components only horizontal smoothing difference error is expected to induce an important negative bias, and this only for some sites (e.g., Thessaloniki, Mainz), while for other sites (e.g., OHP, Xianghe) this is not expected. This means that the bias is at least in some cases due to error in the satellite and/or MAX-DOAS measurement, and not due to comparison error.

We present in the same figure boxplots of the tropospheric $NO_2$ VCD difference between OMNO2 data with QA4ECV MAX-DOAS measurements (blue boxplots). The bias of OMNO2 vs. QA4ECV MAX-DOAS is comparable to that of QA4ECV OMI $NO_2$ vs. QA4ECV MAX-DOAS, although slightly more negative at all sites except Cabauw. If one considers only the subset of OMNO2 pixels where QA4ECV OMI has an accepted pixel, the OMNO2 bias becomes closer to that of QA4ECV OMI for most sites. Although bePRO MAX-DOAS has in principle a better correction for aerosols and vertical profile effects
compared to QA4ECV MAX-DOAS, the bias of QA4ECV OMI with respect to bePRO MAX-DOAS (Fig. 9, green boxes, only for the sites Bujumbura, Uccle and Xianghe) is comparable to that of QA4ECV OMI vs. QA4ECV MAX-DOAS.

We conclude that in most cases, mutual differences between the tropospheric $NO_2$ VCD of the two OMI satellite data products on one hand, and between both MAX-DOAS processings on the other hand, are small compared to the differences between the satellite OMI data products and the MAX-DOAS measurements. The main exception is at the site OHP, where the
median difference and relative difference of OMNO2 vs. QA4ECV MAX-DOAS (-1.4 $\mathrm{Pmolec\,cm^{-2}}$, -60%) is considerably more negative than that of QA4ECV OMI vs. QA4ECV MAX-DOAS (-0.8 $\mathrm{Pmolec\,cm^{-2}}$, -30%). The observation of higher MAX-DOAS tropospheric VCD compared to satellite is a common finding in the literature (e.g., Ma et al., 2013; Kanaya et al., 2014; Chan et al., 2015; Jin et al., 2016; Drosoglou et al., 2017, 2018). The negative bias is therefore not specific to a particular satellite or MAX-DOAS data product.

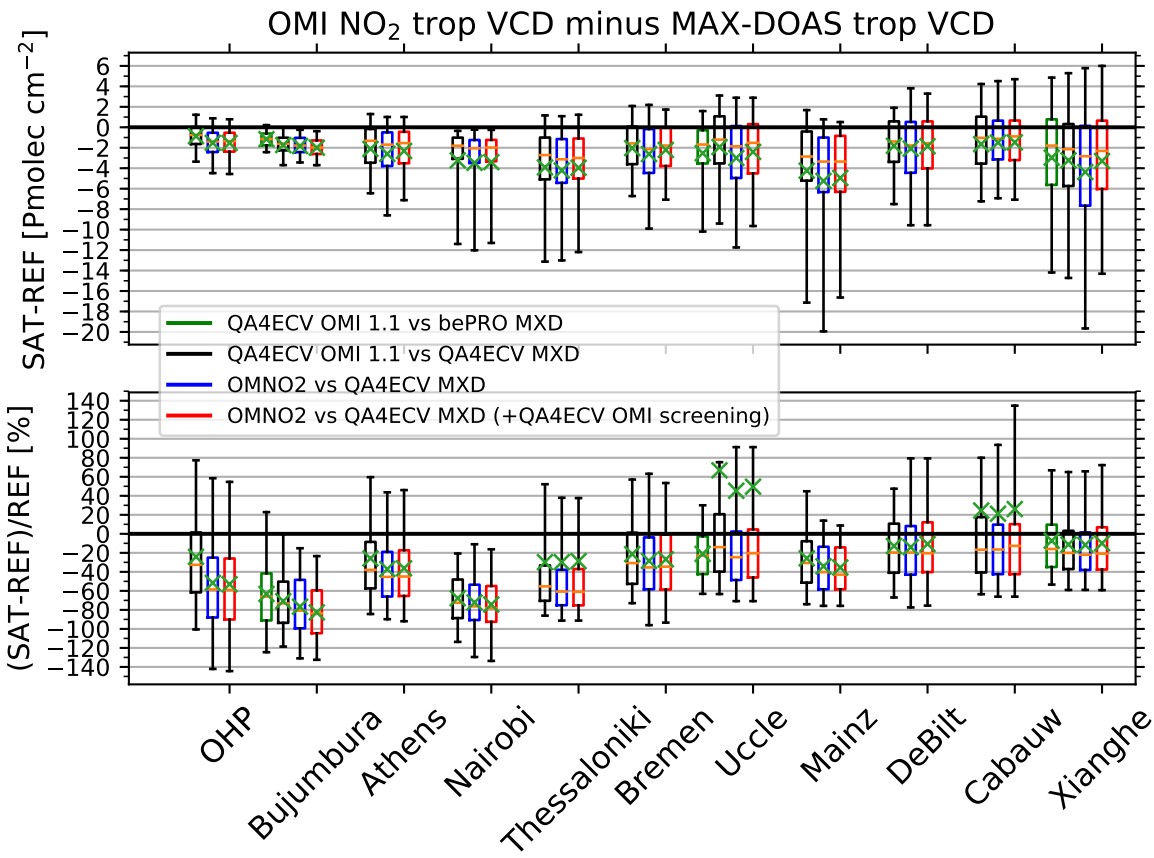

**Figure 9.** Per site, boxplots of QA4ECV OMI $NO_2$ vs. QA4ECV MAX-DOAS (black boxes), QA4ECV OMI $NO_2$ vs. bePRO MAX-DOAS (green boxes, only for 3 sites), OMNO2 vs. QA4ECV MAX-DOAS (blue) and OMNO2 vs. QA4ECV MAX-DOAS, for the subset of OMNO2 pixels where QA4ECV OMI has an accepted pixel (red boxes). The top plot displays boxplots of SAT-REF tropospheric VCD differences, the bottom plot of (SAT-REF)/REF. The same boxplot conventions as in Fig. 3 are applied. Outlying mean relative differences (green crosses) can occur when low REF values are present.



**Seasonal cycle of the bias.** Fig. 10 and 11 present for each site a seasonal plot (i.e., all data mapped to 1 year) of QA4ECV OMI tropospheric $NO_2$ VCD, of QA4ECV MAX-DOAS, and of the difference. Also indicated are rolling monthly mean and median, and outliers identified by iterative 4-$\sigma$ clipping.

A seasonal cycle in the bias, with a larger underestimation in seasons with high $NO_2$, is a recurring feature (Fig. 10). This is the case at the more polluted sites e.g., Xianghe, Mainz, Thessaloniki, in winter months, and is in agreement with several literature results (Ma et al., 2013; Kanaya et al., 2014; Jin et al., 2016). Note however that we find also in the relatively clean site OHP a seasonal cycle in the bias. A very strong seasonal cycle in bias (10-fold increase) is present at Nairobi, where the MAX-DOAS sensor measures a strongly elevated $NO_2$, peaking in July and August, which is not or hardly picked up by the satellite. Likely this is a spatially local phenomenon; this would be consistent with the locally enhanced $NO_2$ in Fig. S5. This site is characterized by local traffic. The enhanced $NO_2$ concentrations in July and August (as measured by MAX-DOAS) are possibly related to meteorology. This season is characterized by low precipitations, low wind speeds (see https://weather-and-climate.com/average-monthly-Rainfall-Temperature-Sunshine,Nairobi,Kenya), and a high cloud cover (as indicated by QA4ECV OMI cloud fraction measurements) limiting $NO_2$ photolysis, therefore a build-up of locally emitted $NO_2$ is a possible explanation. The fact that OMI hardly measures this elevated $NO_2$ can be due to the local character of the emissions.

**Overall discrepancy and consistency with ex-ante uncertainty.** The discrepancy, as measured by the root-mean squared difference (RMSD) between satellite and MAX-DOAS, exceeds for all sites the combined ex-ante uncertainty[6] (see Fig. 12, for the squared quantities). Clearly, comparison error contributes significantly to the RMSD, and/or there are underestimated/unrecognized errors in the satellite or reference data.

The mean squared difference in Fig. 12 is split into 3 additive components: (i) squared mean difference (bias component), (ii) variance of the rolling monthly mean difference (seasonal cycle component) and (iii) variance of the residual difference. The first two components can be attributed to systematic error, the third component to random error and any uncharacterized systematic error. The leading component can be different per site (e.g., bias component at Bujumbura, seasonal component at Nairobi, residual at Mainz and Xianghe).

The satellite and reference data products do not provide the information to split the squared uncertainty according to the random or systematic nature of the error source. Instead, the squared uncertainty in Fig. 12 is separated into additive components according to origin: (i) uncertainty in the MAX-DOAS measurement $u_{GB}$, (ii) uncertainty in the satellite measurement due to error in SCD (expected to be mainly random in nature) $u_{SAT,N_s}$, (iii) stratospheric SCD $u_{SAT,N_{s,strat}}$, and (iv) uncertainty in satellite measurement due to error in tropospheric AMF $u_{SAT,M_{trop}}$. For the sites with the lowest $NO_2$ levels (OHP and Bujumbura), uncertainty in SCD is the main contributor, while for the other sites the MAX-DOAS uncertainty becomes the leading component.

---

[6]Although root-mean squared error (RMSE) and uncertainty are not exactly equivalent, they should be roughly comparable if all error sources are well characterized. If all error is purely random, the RMSE equals the standard deviation of errors, of which the uncertainty is an ex-ante estimate. If the error is fully systematic and constant, RMSE equals the absolute value of the bias, which is expected to be smaller than twice the uncertainty with 95% probability.





**Figure 10.** Seasonal cycle plots for the sites OHP, Bujumbura, Athens, Nairobi, Thessaloniki and Bremen. Top panel: tropospheric VCD of QA4ECV OMI NO$_2$ and QA4ECV MAX-DOAS, and rolling monthly mean and median of both. Bottom panel: differences between QA4ECV OMI NO$_2$ and QA4ECV MAX-DOAS, outliers indicated by 4-$\sigma$ clipping, and rolling monthly mean and median of the difference.

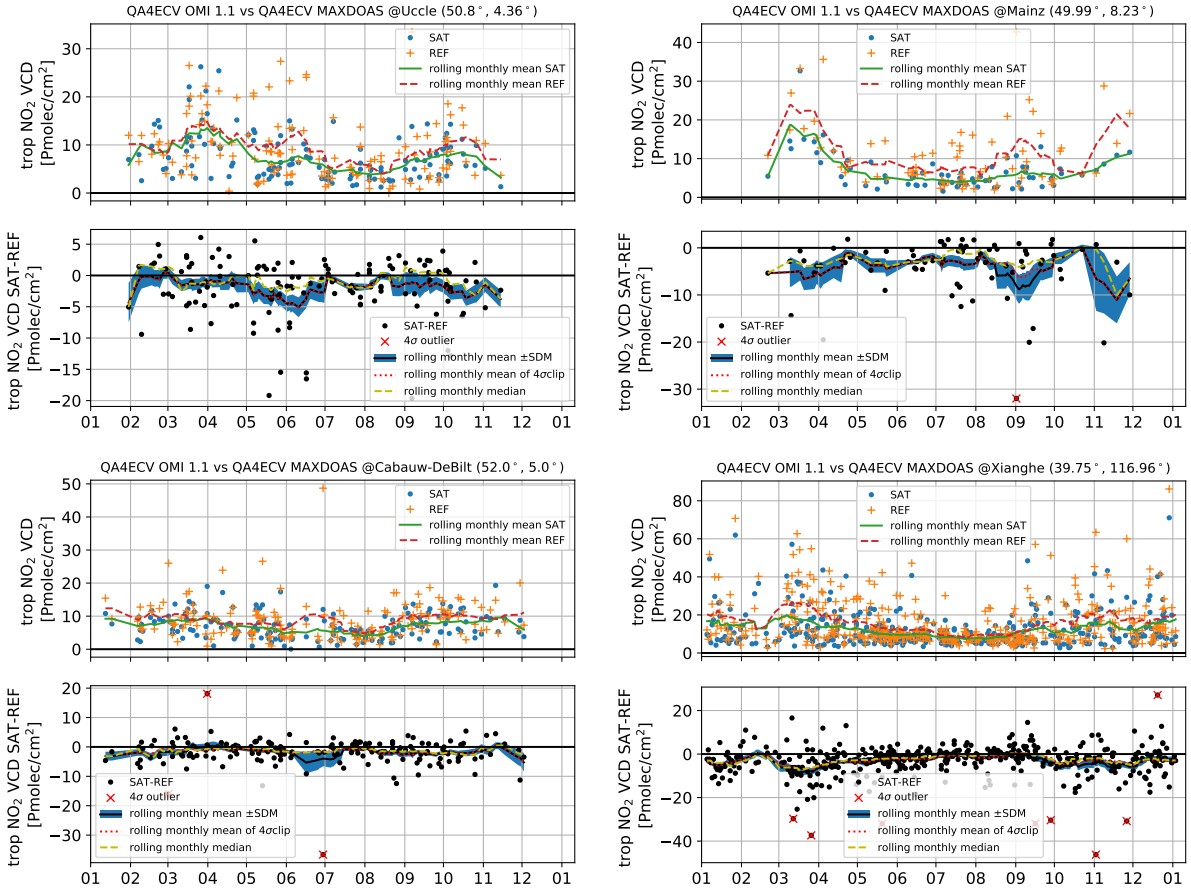

**Figure 11.** As in Fig. 10 but for the sites Uccle, Mainz, Cabauw/De Bilt and Xianghe.

By analysing and intercomparing the tropospheric AMF calculation methods between different retrieval groups, Lorente et al. (2017) concluded that the uncertainty due to differences in retrieval methodology (i.e., methodological uncertainty, termed structural uncertainty by Lorente et al. (2017)) is 32% in unpolluted and 42% in polluted conditions, mostly due to different choices in the ancillary data surface albedo, a priori profile and cloud parameters by different groups. In Fig. 12, this AMF-component of methodological uncertainty, $u_{\mathrm{SAT,meth},M_{\mathrm{trop}}}$, is presented as alternative to the ex-ante $u_{\mathrm{SAT},M_{\mathrm{trop}}}$ obtained by uncertainty propagation, where we classified OHP and Bujumbura as non-polluted sites and the others as polluted. In all cases, the methodological uncertainty exceeds the ex-ante uncertainty $u_{\mathrm{SAT},M_{\mathrm{trop}}}$. At 4 sites, using this methodological uncertainty the discrepancy between OMI and MAX-DOAS can be explained for the most part (Uccle, Cabauw/De Bilt) or even completely (Xianghe), but not at the other sites.

**Modifying screening criteria.** Applying a more strict screening protocol can, at least in principle, mitigate discrepancies in bias and dispersion, at the expense of data loss. In the case at hand, results are mixed for the different sites (see Fig. S9-

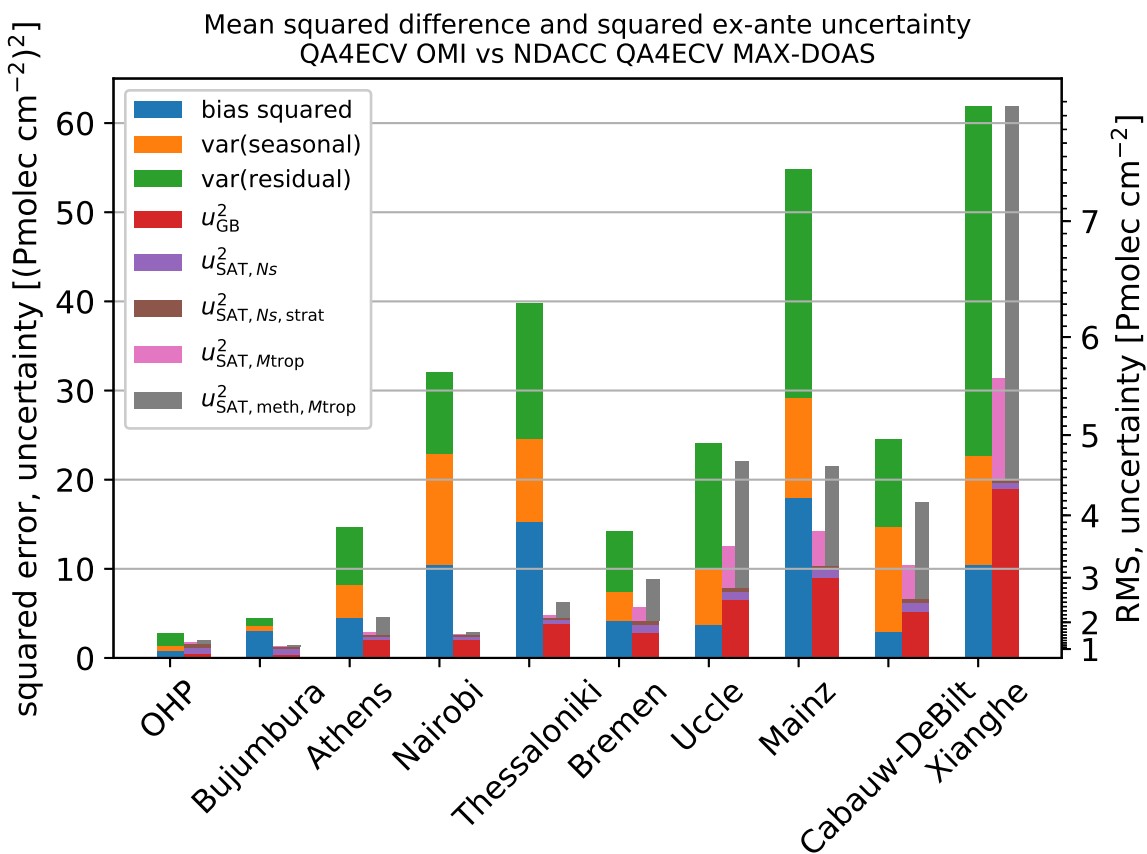

**Figure 12.** Per site two stacked bar plots are provided. The left bar shows the mean squared difference of QA4ECV OMI $NO_2$ vs. QA4ECV MAX-DOAS, split into 3 components: (i) square of the mean difference; (ii) variance of the rolling monthly mean difference; (iii) variance of the residual difference. The right bar shows the combined ex-ante uncertainties of QA4ECV OMI $NO_2$ - QA4ECV MAX-DOAS, split into 4 components: (i) MAX-DOAS squared uncertainty; (ii) QA4ECV OMI squared uncertainty contribution from the total SCD; (iii) from the stratospheric SCD; (iv) QA4ECV OMI squared uncertainty contribution from the tropospheric AMF. Also shown is the AMF-component of methodological uncertainty, derived by intercomparing retrieval methodologies by Lorente et al. (2017, called structural uncertainty in this work). The right y-axis provides a square-root scaling of the corresponding RMS.





S13); stricter criteria does not resolve bias or dispersion for all sites. For the sites Uccle, Mainz, Cabauw and Xianghe strong reductions in bias and/or dispersion ($\sim$0.5-2 $\mathrm{Pmolec\,cm^{-2}}$) can be achieved by filtering more strictly on the effective cloud properties cloud fraction, cloud pressure, the uncertainty component due to cloud pressure $u_{\mathrm{SAT},\,p_{cl}}$, on the MAX-DOAS cloud flag (removing scenes with thick or broken clouds) or on AOD. This suggests that part of the discrepancy is caused by clouds

and/or aerosol. More minor reductions in bias and/or dispersion are achieved for the sites Bujumbura, Nairobi, Athens, Bremen and De Bilt.

Screening more strictly on ground pixel area leads to improvements in bias for Mainz and Thessaloniki, confirming (see section 3.4.5) that horizontal smoothing difference error is a component of the bias. Improvements in dispersion are found for Mainz, Thessaloniki, Uccle and Xianghe.

Using a stricter filtering on effective cloud properties, the RMSD can be made consistent with the ex-ante uncertainty for the sites Uccle and Cabauw-De Bilt (results not shown). For Mainz, this can be achieved if furthermore ground pixels larger than 400 $\mathrm{km^2}$ are excluded (keeping only 25% of the data). Finally, we note that at the site OHP, RMSD and uncertainty are consistent in the months from May to and including August (when $\mathrm{NO_2}$ values are low) without the need of stricter filtering.

For most sites, additional screening (within reasonable limits) cannot lower the RMSD sufficiently to match the uncertainty.

Likely some uncertainty components in either OMI or MAX-DOAS data are underestimated, or not included.

While we found that stricter screening using the uncertainty component due to cloud pressure, $u_{\mathrm{SAT},\,p_{cl}}$, often leads to better results, the obtained threshold values are quite low. This indicates that $u_{\mathrm{SAT},\,p_{cl}}$ is underestimated in the satellite data product. As expected, relaxing the cloud fraction filter beyond the baseline can lead to an increase in bias and/or dispersion (see e.g., Bujumbura, Nairobi, Uccle in Figs. S9-S13), motivating the CF$\leq 0.2$ (or almost equivalently CRF$\leq 0.5$) recommendation. On

the other hand, relaxing the AMF ratio filter beyond the baseline has no large impact on the comparison, while further restricting it has sometimes a negative impact (e.g., increase of bias and/or dispersion at Uccle, Xianghe and Cabauw). Therefore, the current baseline recommended lower bound ($\frac{\mathrm{AMF_{trop}}}{\mathrm{AMF_{geo}}} \geq 0.2$) can be replaced by a lower value (e.g., 0.1 or 0.05).

**Vertical smoothing.** The non-uniform vertical sensitivity of the satellite measurement, combined with an approximate a priori profile shape, is a source of error in the satellite measurement. The bePRO MAX-DOAS provides not only column

but also profile shape information (albeit with a limited vertical resolution) and therefore allows to assess this error source separately. Fig. 13 shows, for the sites Uccle and Xianghe, the impact of directly applying Eq. (3) on the bePRO MAX-DOAS profile (after vertical alignment using the method of Zhou et al. (2009)) on the mean squared deviation (MSD), and its bias, seasonal cycle and residual components. While direct smoothing of the MAX-DOAS profile improves the MSD for Uccle, for Xianghe it increases because the seasonal cycle component increases.

However, one should take into account that the retrieved bePRO profiles have a low vertical resolution and depend on their own a priori profile shape. As is well known (Eq. (10) of Rodgers and Connor (2003), see also the general profile harmonization overview of Keppens et al. (2019)), a priori profiles of satellite and reference should be harmonized before comparison and smoothing. Here, we aligned the surface levels of the profiles following Zhou et al. (2009) and changed the a priori shape profile of the bePRO data to that of the satellite, while keeping the bePRO a priori VCD size (which is actually

obtained from measurement, see Hendrick et al. (2014)) intact. More detail on the applied operations is provided in Section

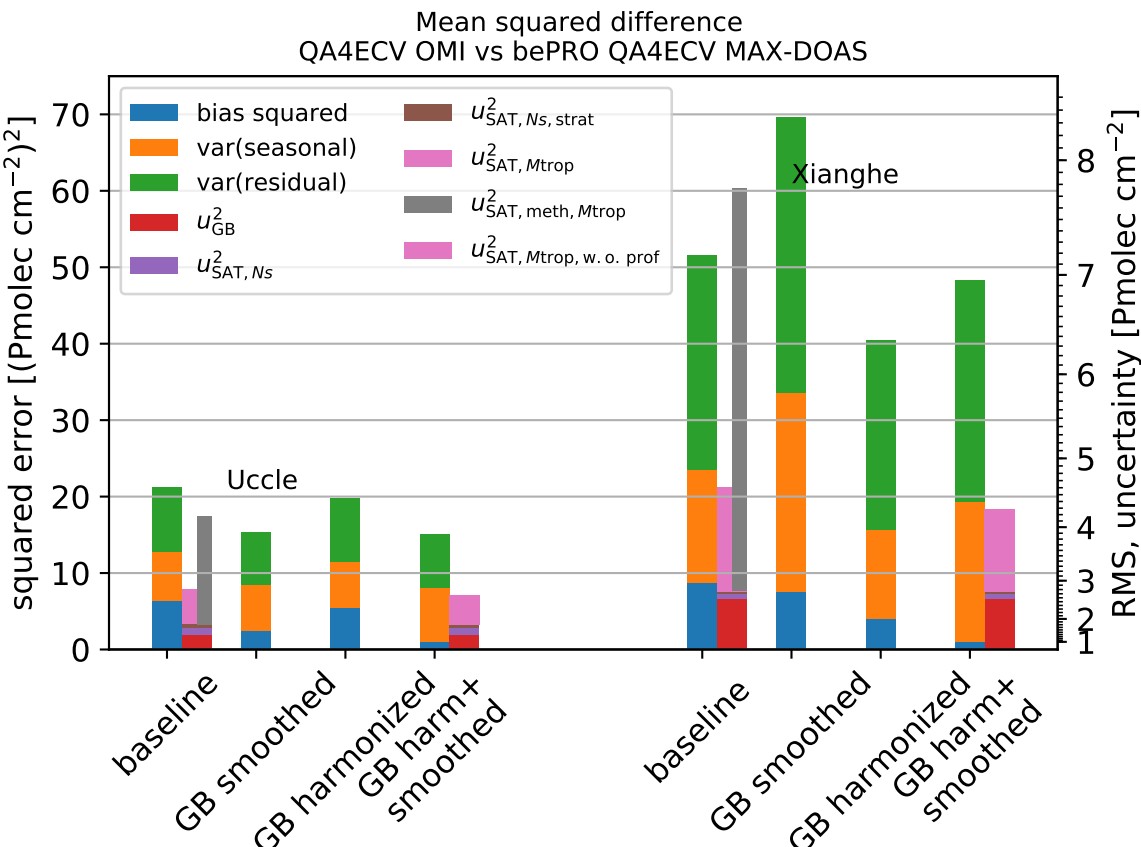

**Figure 13.** Mean squared deviation of QA4ECV OMI vs. bePRO MAX-DOAS tropospheric VCD at Uccle and Xianghe, split in the components squared mean difference (blue), variance of the rolling monthly mean difference (orange) and variance of the residual difference (green). Per site, from left to right: (i) baseline comparison, (ii) MAX-DOAS profile smoothed by the OMI averaging kernel, (iii) MAX-DOAS a priori replaced with that of the satellite, (iv) First a priori harmonization, then smoothing. Details of the operations are provided in S6. At the baseline (i), also the squared ex-ante uncertainty (divided into components) is provided. The same squared ex-ante uncertainty, minus the satellite profile shape uncertainty contribution, is provided at (iv).





S6. The harmonization operation reduces all components of the MSD (bias, seasonal cycle and residual component) for the Xianghe site. Applying in addition smoothing after the a priori harmonization, the bias component (blue bar in Fig. 13) is almost completely removed, but the other two components increase. Application of the averaging kernel therefore does not lead necessarily to an improvement in all aspects of a comparison; this should be the focus of further research. As the bias

component is removed almost completely after harmonization and smoothing at Uccle and Xianghe, one can conclude that -at least at these two sites- the bias is largely due to errors in the a priori profile shape. Using better quality a priori profiles in both satellite and MAX-DOAS data (e.g., from regional scale models) is therefore recommended.

When the averaging kernel is applied, it is recommended to remove the satellite a priori shape component from the uncertainty budget (Boersma et al., 2018). This component was tentatively assigned 10% of the VCD value. This only leads to a

modest reduction of the combined uncertainty in Fig. 13 (compare the non-hatched and hatched pink bars) as the dominant contribution to the OMI AMF uncertainty component is related to surface albedo rather than profile shape. E.g., the combined uncertainty at Uccle reduces from 2.8 to 2.7 $\mathrm{Pmolec\,cm^{-2}}$. However, the smoothing operation reduces the RMSD at Uccle by about 2 $\mathrm{Pmolec\,cm^{-2}}$, strongly suggesting that the current 10% uncertainty assignment is an underestimate.

## 4   Conclusion

In this work, stratospheric and tropospheric $NO_2$ VCD of the QA4ECV OMI 1.1 data product are validated, using ground-based NDACC ZSL-DOAS data and MAX-DOAS data, respectively. Two MAX-DOAS processings are used, the NDACC bePRO profile retrieval and the harmonized QA4ECV column retrieval.

Quality screening according to the data product provider recommendations is an essential step before the satellite product can be used. However, a user (e.g., a developer of L3-type temporally averaged data) should be aware that for tropospheric

VCD this leads to a preference of cloud-free scenes and therefore to a negative sampling bias especially at polluted sites (strong reduction in mean VCD from 24 to 15 $\mathrm{Pmolec\,cm^{-2}}$ at Xianghe, and reduction by a few $\mathrm{Pmolec\,cm^{-2}}$ at Nairobi, Bremen, Thessaloniki and De Bilt/Cabauw). This sampling bias is reduced at De Bilt and Bremen by relaxing the lower bound filter on $\frac{M_{\mathrm{trop}}}{M_{\mathrm{geo}}}$ from 0.2 to 0.05.

The QA4ECV OMI stratospheric $NO_2$ VCD has a small (mostly wintertime) bias with respect to the ZSL-DOAS measure-

ments of the order of $-0.2\pm0.06\,\mathrm{Pmolec\,cm^{-2}}$ (5-10%) and a dispersion of 0.2 to 1 $\mathrm{Pmolec\,cm^{-2}}$, with a good representation of the seasonal cycle.

QA4ECV OMI tropospheric $NO_2$ VCD is negatively biased vs. the MAX-DOAS data. This is not unique to this data product; the same conclusion is reached for NASA's OMI OMNO2 data product, and for several other tropospheric $NO_2$ data products in the literature. The overall discrepancy exceeds the combined ex-ante uncertainty of satellite and MAX-DOAS data. This is

a conclusion opposite to the one of Boersma et al. (2018), where uncertainties seemed overestimated, although that was for a single site in a one-month time period (Tai'an, China, June 2006).

We studied a wide range of potential error sources of the discrepancy in tropospheric VCD between satellite and MAX-DOAS. An overview is provided in Table 3.





**Table 3.** Overview of discrepancies and error sources studied in this work. (MXD=MAX-DOAS)

| Contribution | Description |
|---|---|
| full discrepancy $e_{\text{total}}$ | Negative bias ranging from $-0.9\,\mathrm{Pmolec\,cm^{-2}}$ (OHP) to $-4\,\mathrm{Pmolec\,cm^{-2}}$ (Mainz, Thessaloniki). RMSD ranging from 2 (OHP, Bujumbura) to $8\,\mathrm{Pmolec\,cm^{-2}}$ (Xianghe). RMSD dominated by bias in Bujumbura and Thessaloniki, by seasonal cycle dispersion in Nairobi and by residual dispersion otherwise. |
| *Comparison errors* | |
| temporal sampling diff. error $e_{\Delta t}$ | Mitigated by averaging MXD within 1h interval. No systematic component. Impact on dispersion[1]: $\leq 0.1\,\mathrm{Pmolec\,cm^{-2}}$ (low pollution) to 0.1 to $0.5\,\mathrm{Pmolec\,cm^{-2}}$ (high pollution). |
| horizontal sampling diff. error $e_{\Delta r}$ | Mitigated by excluding ground pixels not covering site. Systematic component between zero and $-0.6\,\mathrm{Pmolec\,cm^{-2}}$. Impact on dispersion[1]: $\leq 0.1\,\mathrm{Pmolec\,cm^{-2}}$ (low pollution) to $\leq 0.6\,\mathrm{Pmolec\,cm^{-2}}$ (high pollution). |
| vertical sampling diff. error $e_{\Delta z}$, surface level | Alignment of satellite a priori profile to MXD surface level using the method of Zhou changes bias by $\leq 0.3\,\mathrm{Pmolec\,cm^{-2}}$. Bujumbura: complicated oreography might lead to a higher bias. Athens: MXD on hill is a likely source of positive bias |
| vertical sampling diff. error $e_{\Delta z}$, top grid level | MAX-DOAS VCD restricted to lower troposphere. Correction estimated from satellite upper tropospheric a priori profile increases the bias. |
| horizontal smoothing diff. error $e_{Sr}$ | Qualitatively assessed. Contributes to bias in Nairobi, Thessaloniki, Mainz, and does not contribute (significantly) to bias in OHP, Cabauw and Xianghe. For other sites the results are mixed. |
| *Measurement/retrieval errors* | |
| OMI total SCD error $e_{\text{SAT},s}$ | Impact of noise term on dispersion[1]: $\leq 0.1\,\mathrm{Pmolec\,cm^{-2}}$ (low pollution), negligible (high pollution). |
| OMI strat. SCD error $e_{\text{SAT},s,\text{strat}}$ | Bias in strat VCD of $\sim -0.2\,\mathrm{Pmolec\,cm^{-2}}$ translates (via $\frac{M_{\text{strat}}}{M_{\text{trop}}}$) to $\sim+0.6\,\mathrm{Pmolec\,cm^{-2}}$ in trop VCD. |
| OMI trop. AMF error $e_{\text{SAT},M_{\text{trop}}}$ | 32% to 42% (Lorente et al., 2017), dominated by choice in a priori profile, cloud parameters and surface albedo. This could explain (most or all) of the discrepancy in Uccle, Cabauw/De Bilt and Xianghe. |
| error due to cloud or aerosol (OMI or MXD) | Strong reduction in bias and/or dispersion by stricter filtering, for Uccle, Mainz, Cabauw and Xianghe. Simulations (Ma et al., 2013; Jin et al., 2016) indicate cloud or aerosol can cause a factor 2 underestimation for satellite and up to 20% overestimation for MXD. |
| error due to vertical smoothing | Only assessed with bePRO MXD at Uccle and Xianghe. Applying a priori harmonization and smoothing. Mean difference reduces from -3 to $-1\,\mathrm{Pmolec\,cm^{-2}}$, and median difference from -2 to $0\,\mathrm{Pmolec\,cm^{-2}}$. RMSD: small reduction. |

[1]. 'Impact on dispersion': stated is by how much would the standard deviation of $N_{v,\text{trop,SAT}} - N_{v,\text{trop,REF}}$ reduce if the estimated standard deviation due to this particular error source is subtracted in quadrature.





At several sites the MAX-DOAS instrument is located close (within satellite pixel distance) to an emission source and therefore horizontal smoothing difference error explains (part of) the bias, but there are also a few cases (OHP, Cabauw, Xianghe) where this does not hold. Sampling difference errors were found to be either minor (temporal, horizontal), or to contribute in the opposite direction (vertical).

Measurement/retrieval error in satellite and MAX-DOAS data are other potential sources of discrepancy. Errors in satellite total SCD and stratospheric SCD do not contribute much, leaving errors in satellite tropospheric AMF or MAX-DOAS data as candidate error sources. Part of the discrepancy is caused by errors in either satellite or MAX-DOAS measurement induced by (low) clouds and/or aerosol (e.g., at the sites Mainz, Xianghe). According to radiative transfer simulations (Ma et al., 2013; Jin et al., 2016), these effects impact the satellite tropospheric $NO_2$ VCD measurements (factor ~2 decrease) more than the

MAX-DOAS measurements (overestimation of at most 20%). Also the non-uniform vertical sensitivity of OMI and uncertainty in the a priori profile shape contributes to the discrepancy, as shown here with the QA4ECV OMI vs. bePRO MAX-DOAS comparison. This is in agreement with the work of Lorente et al. (2017): the uncertainty in retrieval method (due to inter-team retrieval setting differences; shorthand methodological uncertainty) in tropospheric AMF is dominated by differences in a priori profile, cloud parameters and surface albedo. Moreover, using this uncertainty estimate for the AMF instead of the ex-ante,

one can explain the SAT-REF tropospheric VCD discrepancies for 3 sites (Uccle, Cabauw/De Bilt, Xianghe). For these 3 sites, consistency can also be reached by filtering more strictly on cloud parameters.

    Finally, for some of the discrepancies there is no straightforward explanation. A first example is the negative bias at OHP in winter time. Possibly, this is related to a lower tropospheric AMF in winter time, as the planetary boundary layer is more shallow and the SZA is higher. As a result, comparisons become more sensitive to e.g., errors in the profile shape. A second

example of unexplained discrepancy is the negative bias at Nairobi, even when focusing on the months December-March when MAX-DOAS measured tropospheric VCD values are relatively low.

    The potential impact of horizontal smoothing difference error was analyzed in this work in a rather qualitative way. Analysis using Observing System Simulation Experiments at fine spatial resolution (Verhoelst et al., 2015), or other experimental set-ups (e.g., sensors measuring in multiple azimuth directions (Brinksma et al., 2008; Ortega et al., 2015)), can improve on this.

The inter-team harmonization of MAX-DOAS data within the QA4ECV project is an important step forward for satellite validation, although some issues remain e.g., regarding the harmonisation of reported uncertainties. The ESA-funded project FRM4DOAS (http://frm4doas.aeronomie.be) should improve on this with the development of a first central processing system for MAX-DOAS measurements built on state-of-the-art retrieval algorithms and corresponding settings.

    The availability of an ex-ante uncertainty per measurement, and its decomposition in source components, greatly facilitates

the validation. However, information on how individual measurement uncertainties should be combined is incomplete in the satellite and MAX-DOAS data files. This limits the ability to check if e.g., the overall bias, the dispersion, or seasonal cycle of the bias each separately are within expectations; in this work we only checked consistency of the overall discrepancy (expressed as RMSD) with the combined total uncertainty. It is recommended that information on the systematic/random nature and error correlation is included in the satellite data product.





The ex-ante per-pixel uncertainty in the QA4ECV $NO_2$ satellite data product is likely underestimated. A solution could be to explicitly account for the methodological uncertainty on AMF; similar as done for the QA4ECV HCHO data product (De Smedt et al., 2018). Alternatively, the uncertainty component due to profile shape in the OMI product could be increased, as tests in this work show that the current 10% assignment is an underestimate. The QA4ECV $NO_2$ recommended filter on

AMF ratio can be made less restrictive (e.g., 0.05 lower bound), reducing data loss and sampling bias without compromising the comparisons with MAX-DOAS. Furthermore, replacement of the coarsely resolved TM5 $NO_2$ profiles with high-spatial resolution profiles from regional air-quality analyses (e.g., CAMS regional, http://www.regional.atmosphere.copernicus.eu) would be very helpful to bridge part of the gap between MAX-DOAS and OMI.

*Code and data availability.* The QA4ECV OMI $NO_2$ data is available via http://www.qa4ecv.eu, under "ECV data". The OMNO2 data is

10 publicly available from the NASA Goddard Earth Sciences (GES) Data and Information Services Center public website: https://disc.gsfc. nasa.gov/datasets/OMNO2_V003/summary/. The ZSL-DOAS data and bePRO MAX-DOAS as part of the Network for the Detection of Atmospheric Composition Change (NDACC) are publicly available (see http://www.ndacc.org). The QA4ECV MAX-DOAS data is available at http://uv-vis.aeronomie.be/groundbased/QA4ECV_MAXDOAS/index.php; it is mandatory to contact instrument principal investigators for any usage of the data. The AERONET AOD data is available at http://aeronet.gsfc.nasa.gov. Sentinel-5p $NO_2$ RPRO (reprocessed) and

15 OFFL (offline) data 01.02.00-01.02.02 can be obtained from the Sentinel-5P Pre-Operations Data Hub (https://s5phub.copernicus.eu/dhus/#/home).

Part of the validation processing was performed with the data harmonization toolset HARP (©S[&]T, the Netherlands), available at https://github.com/stcorp/harp under the BSD 3-Clause "New" or "Revised" License.





*Author contributions.* SC coordinated the paper and carried out the validation analysis. TV carried out the stratospheric VCD validation analysis. GP contributed insights on the tropospheric VCD validation analysis. DH, AK, J-CL and TV contributed validation expertise. JG, SN and BR created software tools for validation. JG performed general data collection and format harmonisation. FH coordinated the creation of the QA4ECV MAX-DOAS improved data sets. AB, J-PB, FH, AP, JR, AR, MVR, TW are Principal Investigators for the QA4ECV MAX-DOAS measurements. FH and MVR are Principal Investigators for the bePRO MAX-DOAS measurements. FG, AP, and J-PP are Principal Investigators for the SAOZ ZSL-DOAS measurements. FB, HE, AR, IDS, AL, JVG, EP, MVR and TW are the authors of the QA4ECV $NO_2$ OMI data set. J-CL is the coordinator of this research. All authors reviewed and commented on the manuscript.

*Competing interests.* The authors declare that they have no conflict of interest.

*Acknowledgements.* This research was funded by the EU FP7 project Quality Assurance for Essential Climate Variables (QA4ECV, grant no. 607405) and the EU H2020 project Gap Analysis for Integrated Atmospheric ECV CLImate Monitoring (GAIA-CLIM, Ares(2014)3708963/Project 640276). In particular the generation of harmonized QA4ECV OMI and MAX-DOAS data sets was funded by the EU FP7 project QA4ECV. Several validation analysis tools were funded by the Belgian Science Policy Office (BELSPO) and ESA through the ProDEx-10 project ACROSAT. We are grateful to Marina Zara (KNMI) for clarifications about the different QA4ECV $NO_2$ OMI uncertainty fields. Ground-based ZSL-DOAS data and other MAX-DOAS data used in this publication were obtained as part of the Network for the Detection of Atmospheric Composition Change (NDACC) and are publicly available (see http://www.ndacc.org). NASA OMNO2 data were obtained through NASA's Earth Observing System Data and Information System (EOSDIS). We are grateful to Nickolay A. Krotkov and Lok Nath Lamsal (NASA/GSFC) for clarifications about the NASA OMNO2 data product. We are also grateful to Trissevgeni Stavrakou (BIRA-IASB) for fruitful discussions on tropospheric $NO_2$ chemistry. The European Commission is further acknowledged for having supported cross-fertilisation meetings among FP7 (CLIP-C, QA4ECV, ERACLIM-2, EUCLEIA, EUPORIAS, UERRA) and H2020 (GAIA-CLIM, FIDUCEO) climate service related projects. Regarding the AERONET data, we thank the Principal Investigators Rachel Akimana, Vassilis Amiridis, Meinrat Andreae, Alkiviadis Bais, Philippe Goloub, J.S. Bas Henzing, Christian Hermans, Eughne Ndenzako, Pierre Nzohab-onayo, Michel Van Roozendael, Ucai Wang, Xiangao Xia, and their staff for establishing and maintaining the 8 AERONET sites used in this investigation. Sentinel-5 Precursor $NO_2$ data has been used in this work. Sentinel-5 Precursor is a European Space Agency (ESA) mission on behalf of the European Commission (EC). The TROPOMI payload is a joint development by ESA and the Netherlands Space Office (NSO). The Sentinel-5 Precursor ground-segment development has been funded by ESA and with national contributions from The Netherlands, Germany, and Belgium.



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
