# Peer review of "Validation of Aura-OMI QA4ECV NO2 Climate Data Records with ground-based DOAS networks: role of measurement and comparison uncertainties"

_Atmospheric Chemistry and Physics, 2019_

## Referee Comment (RC1) · Anonymous Referee #1 · 23 Jan 2020

General comments:

The manuscript by Compernolle et al. presents an important validation work for a satellite-based NO2 climate data records. The validation process is accurate and comprehensive. The findings, especially the ex-ante uncertainty vs. RMSD budget, in this work are important for end users. The manuscript should be published after addressing the following minor comments.

Specific comments:

[Figure]

Page 10, lines 1 to 8. Any comments on the difference caused by AOD inputs (i.e., QA4ECV uses AERONET, whereas bePRO uses OE results)? What if bePRO uses AERONET AOD?

Page 17, Figure 5. The model adjusted SAOZ AM and PM data have a larger discrepancy in winter at OHP (Fig. 5 right panel, green and cyan dash lines). This effect is not observed at Kerguelen, i.e., its July data. Is this due to the heavy local tropospheric $NO_2$ signal at OHP (in winter)? Any comments?

Page 20, Figure 7. It is very difficult to see if there is or isn't any seasonal variation. The symbols are jammed. One needs to find better methods to show this, e.g., sub-panels by seasons.

Page 24, Figure 8. Taking the fact the size of superpixels from OMI and TROPOMI are similar, why Uccle and Thessaloniki show very large changes in the smoothing difference error (OMI vs. TROPOMI). Is this indicate these sites have more fine-scale variation than others? Note that for Mainz, the smoothing difference error in Fig. 8a and 8b are not very different.

Figure 8. Any comments on the positive mean difference for Xianghe and Cabauw in Fig 8b, at JFM? This figure is fascinating and revelled many important aspects of these two generations of satellite data products. Although this is a bit out of the scope for this paper, I would still suggest the author give more comments on their difference.

Page 31, lines 28-29. For the Xianghe site, why smoothing can increase the seasonal variance this much (baseline vs. GB smoothed, var (seasonal) increased by about a factor of two)? Can the author confirm this is simply due to the non-harmonized a priori? Why "GB harmonized" has less seasonal variance than "GB harm + smoothed". The figure leaves the impression that the averaging kernel smoothing caused this increase of seasonal variance, no matter one harmonizes a priori or not. Please provide some comments and explanations.

Technical corrections:

Page 3, line 8. Define OMI here.

Page 3, line 9. Define DOAS here.

Fig. 4. Use consistent abbreviations for stratospheric and tropospheric, i.e., "strat" or "strato", "trop" or "tropo".

Page 8, line 26. Modify "by Irie et al. (e.g., 2011, Fig. 17)" to "by e.g., Fig. 17 in Irie et al. 2011."

Page 10, line 2. AOD has been defined twice in this line, remove the 2nd one.

Page 11, line 21. Change 'MAXDOAS' to 'MAX-DOAS'.

Page 15, Figure 3. Please define "MXD" in the caption.

Page 38, line 7. Change "n/a-n/a" to proper page numbers. There are several other "n/a-n/a" in the references.
* * *

---

## Referee Comment (RC2) · Anonymous Referee #2 · 30 Jan 2020

**General Comments**

The manuscript entitled 'Validation of Aura-OMI QA4ECV NO$_2$ Climate Data Records with ground-based DOAS networks: role of measurement and comparison uncertainties' by Compernolle et al. describes the results of a validation exercise, comparing satellite-borne QA4ECV tropospheric and stratospheric NO$_2$ partial VCDs with ground-based observations from a large number of stations.

The paper is very well written and represents a significant contribution to the validation

of satellite observations. Data products, validation methodology and data screening are described in detail. Error sources and potential reasons for discrepancies between ground-based and satellite-borne observations are discussed thoroughly. As far as I can judge as a non-native English speaker, there are hardly any grammatical or syntactical errors. I recommend the publication after addressing some minor issues as listed below. In particular, I would appreciate if the processing of the ground-based data sets and the differences between QA4ECV and bePRO data products would be discussed in some more detail.

**Specific Comments**

Section 2.2.2: I feel that the MAX-DOAS retrieval algorithms should be described in more detail. It should be stated more clearly that the QA4ECV and the bePRO algorithms are distinctly different, with QA4ECD retrieving NO2 VCDs directly by dividing the dSCD from a single elevation angle by the differential AMF, while bePRO VCDs are determined by integrating a vertical NO2 profile retrieved by an OEM algorithm based on measurements from several elevation angles.

There are recent studies on the performance of bePRO in comparison with other profile retrieval algorithms (Frieß et al., Atmos. Meas. Tech., 2019, https://doi.org/10.5194/amt-12-2155-2019; Tirpitz et al., Atmos. Meas. Tech. Discuss, https://doi.org/10.5194/amt-2019-456) which should be cited here. I would furthermore appreciate if it would be discussed to what extent the problems with the stability of the bePRO NO2 vertical profile retrieval identified within these studies affects the quality of the data used here for OMI validation.

P7, L10: Please provide a reference (or an URL) for the description of the NDACC standard procedure.

P8, L31: Given that QA4ECV MAX-DOAS tropospheric NO2 is determined by dividing the tropospheric SCD by the tropospheric AMF, I don't understand how a vertical grid can be involved here.

P12, L6: Explain what you mean with the term 'observation operator'.

P12, L7: Which ray tracing code did you use here?

Last paragraph of Section 3.3 and Figure 7: It is not clear to me in which way the 'bias-correction for the annual mean difference' has been performed - please explain in more detail.

P23, L6: By how much has the satellite a priori profile been shifted?

**Technical Comments**

P21, L4: 'characterized with' -> 'characterized by'

P21, L22: Insert 'being' before 'tropospheric VCD'

P22, L25: Add right parenthesis: '(see Eq. (1))'

P23, L12 and L30: Add space between number and unit

P31, L1: 'does' -> 'do'

---

## Author Response (AR1)

**Changes to the manuscript: "Validation of Aura-OMI QA4ECV NO$_2$ Climate Data Records with ground-based DOAS networks: role of measurement and comparison uncertainties"**

S. Compernolle, T. Verhoelst, et al.

April 30, 2020

**Abstract**

Note: we refer here to pages and line numbers of the marked-up manuscript version (obtained using latexdiff).

**Answers to reviewer 1 and corresponding changes**

**Reviewer 1:** *The manuscript by Compernolle et al. presents an important validation work for a satellite-based NO2 climate data records. The validation process is accurate and comprehensive. The findings, especially the ex-ante uncertainty vs. RMSD budget, in this work are important for end users. The manuscript should be published after addressing the following minor comments.*

**Author reply 1:** We thank reviewer 1 for his positive comments and helpful remarks. We give a point-by-point answer below.

**Reviewer 1:** *Page 10, lines 1 to 8. Any comments on the difference caused by AOD inputs (i.e., QA4ECV uses AERONET, whereas bePRO uses OE results)? What if bePRO uses AERONET AOD?*

**Authors reply 2:** One obvious limitation of using the AERONET AOD with bePRO NO2 is a loss of data: for part of the bePRO NO2 data no co-located AERONET AOD is available. This was also a limitation for the QA4ECV MAX-DOAS NO2 + AERONET AOD combination (see the dashed purple line in Fig. S9 and following). Also, as opposed to the bePRO NO2/bePRO AOD, co-located bePRO NO2/AERONET AOD pairs have a temporal co-location mismatch and (where instruments are at different locations) spatial co-location mismatch. Therefore, conclusions based on bePRO NO2/AERONET AOD combination are in generally less clear than for bePRO NO2/bePRO AOD.

**Manuscript changes:**

Page 9, line 12. Regarding bePRO, added "measurements at the same temporal sampling as the NO2 measurements."

Page 10, line 7. New text: A limitation when investigating AOD dependencies in satellite-MAX-DOAS comparisons using AERONET AOD with QA4ECV MAX-DOAS tropospheric $NO_2$ VCD data (as compared to using bePRO AOD with bePRO $NO_2$ data) is that it implies a subsetting: for part of the QA4ECV MAX-DOAS $NO_2$ data no co-located AERONET AOD is available. Also, as opposed to the bePRO $NO_2$/bePRO AOD combination, co-located QA4ECV MAX-DOAS $NO_2$/AERONET AOD data pairs have a temporal co-location mismatch and (where instruments are at different locations) a spatial co-location mismatch. A test was performed (results not shown) using the bePRO $NO_2$/AERONET AOD combination. It was generally found that the results are less clear than for the bePRO $NO_2$/bePRO AOD combination.

**Reviewer 1:** *Page 17, Figure 5. The model adjusted SAOZ AM and PM data have a larger discrepancy in winter at OHP (Fig. 5 right panel, green and cyan dash lines). This effect is not observed at Kerguelen, i.e., its July data. Is this due to the heavy local tropospheric NO2 signal at OHP (in winter)? Any comments?*

**Authors reply 3:** Weve added a paragraph on this, but it remains an open question. The tropospheric column is in principle the one above the station for both sunrise and sunset observations (as it is below the scattering altitude), so it would require a diurnal cycle in the tropospheric column at the station to cause this discrepancy. This could be investigated with MAX-DOAS data, but that is beyond the scope of the current analysis.

**Manuscript changes:** Page 18, line 8. New paragraph. At OHP, the wintertime agreement between sunrise and sunset after photochemical adjustment is not as good. Contamination by tropospheric pollution is expected to be similar for both sunrise and sunset measurements, as it contributes to the airmass below the scattering altitude, i.e. the column above the station, as opposed to the large and offset area of sensitivity in the stratosphere. Differences between sunrise and sunset contamination could still be caused by a diurnal cycle in the tropospheric column, but an analysis of that diurnal cycle (e.g. from MAX-DOAS data) is beyond the scope of this work.

**Reviewer 1:** *Page 20, Figure 7. It is very difficult to see if there is or isnt any seasonal variation. The symbols are jammed. One needs to find better methods to show this, e.g., sub-panels by seasons.*

**Authors reply 4:** We chose an entirely different visualization in the new version of the manuscript, which is hopefully clearer, and actually contains more information.

**Manuscript changes:** Page 17, Figure 6. Here is now a mosaic plot included (x:Day of Year, y: latitude, z: median difference)

**Reviewer 1:** *Page 24, Figure 8. Taking the fact the size of superpixels*

*from OMI and TROPOMI are similar, why Uccle and Thessaloniki show very large changes in the smoothing difference error (OMI vs. TROPOMI). Is this indicate these sites have more fine-scale variation than others? Note that for Mainz, the smoothing difference error in Fig. 8a and 8b are not very different.*

**Authors reply 5:** There are two major differences between the Fig. 8a and Fig. 8b approach: (i) the different size of the central pixel, which is way bigger for OMI. As the reviewer suggests therefore, one expects more sensitivity to fine-scale variation in Fig. 8b. (ii) the different temporal range (starting from 2004 for OMI and from 2018 for TROPOMI) will capture differently evolution in NO2 concentration patterns (caused by e.g., new emission policies over time). We added a sentence citing the above two differences in approach as probable reasons for differences between Fig. 8a and 8b.

**Manuscript changes:** Page 23, line 6. New text. Differences between the OMI and TROPOMI-based calculations are likely caused by (i) the much larger central pixel of OMI compared to TROPOMI, leading to a lower sensitivity to fine-scale variations in Fig. 8a, and (ii) evolution in e.g., $NO_2$ concentration patterns, captured differently by the different temporal ranges used in Fig. 8a and b.

**Reviewer 1:** *Figure 8. Any comments on the positive mean difference for Xianghe and Cabauw in Fig 8b, at JFM? This figure is fascinating and revelled many important aspects of these two generations of satellite data products. Although this is a bit out of the scope for this paper, I would still suggest the author give more comments on their difference.*

**Authors reply 6:** As opposed to many of the other MAX-DOAS sensors, these two sites are not located at urban centers, although pollution centers are in the neighbourhood. Therefore, the positive mean differences at JFM captured by TROPOMI could well be due to NO2 fields captured at the border of the TROPOMI superpixel. This is in agreement with very recent work of Pinardi et al. (2020) on the horizontal smoothing effect. The estimated horizontal dilution factors in Fig. S3 of Pinardi et al. (2020) are positive for Cabauw and Xianghe, indicating that NO2 is higher in the periphery than at the MAX-DOAS location. Based on the above, we added an explanation in the text as an example of the higher sensitivity to fine-scale variation of the TROPOMI-based calculation. (See the previous remark on the same Figure).

New Reference: Pinardi et al. (2020) Pinardi, G.; Van Roozendael, M.; Hendrick, F.; Theys, N.; Abuhas- san, N.; Bais, A.; Boersma, F.; Cede, A.; Chong, J.; Donner, S.; Drosoglou, T.; Frie, U.; Granville, J.; Herman, J. R.; Eskes, H.; Holla, R.; Hovila, J.; Irie, H.; Kanaya, Y.; Karagkiozidis, D.; Kouremeti, N.; Lambert, J.-C.; Ma, J.; Peters, E.; Piters, A.; Postylyakov, O.; Richter, A.; Remmers, J.; Takashima, H.; Tiefengraber, M.; Valks, P.; Vlemmix, T.; Wagner, T. and Wittrock, F. Validation of tropospheric NO 2 column measurements of GOME-2A and OMI using MAX-DOAS and direct sun network observations, Atmospheric Measurement Techniques Discussions, 2020 , 2020, 1-55 10.5194/amt-2020-76

**Manuscript changes:** Page 23, line 8. New text. A case in point are the positive mean differences in JFM and OND captured in the TROPOMI-based calculation but not in the OMI-based calculation. Both MAX-DOAS sensors are not located at urban centers, although pollution centers are in the neighbourhood. Therefore, the positive mean differences at JFM and OND captured by TROPOMI is likely due to $NO_2$ fields in the periphery of the TROPOMI superpixel. This is in agreement with very recent work of Pinardi et al. (2020) on the horizontal smoothing effect. The estimated 'horizontal dilution factors' in Fig. S3 of Pinardi et al. (2020) are positive for Cabauw and Xianghe, indicating that $NO_2$ is higher in the periphery than at the MAX-DOAS location.

**Reviewer 1:** *Page 31, lines 28-29. For the Xianghe site, why smoothing can in- crease the seasonal variance this much (baseline vs. GB smoothed, var (seasonal) increased by about a factor of two)? Can the author confirm this is simply due to the non-harmonized a priori? Why GB harmonized has less seasonal variance than GB harm + smoothed. The figure leaves the impression that the averaging kernel smooth- ing caused this increase of seasonal variance, no matter one harmonizes a priori or not. Please provide some comments and explanations.*

**Authors reply 7:** The increase in seasonal variance is caused by the interplay of the seasonal variation of the MAX-DOAS vertical profile and of the satellite vertical averaging kernel. Specifically, it is found for the Xianghe case that in wintertime averaging ker- nels have higher values close to the surface while MAX-DOAS NO2 profiles can also be peaked at the surface. The combination causes increased MAX-DOAS columns upon vertical smoothing. This is seen e.g., in the comparison of GOME-2 AC SAF GDP 4.8 NO2 product with MAX-DOAS at Xianghe (see Fig. 7.14 of AC-SAF 2018 report and Figs. S3 and S5 of Liu et al. (2019)) and Figs. S3 and S5 of Liu (2019)). While the a priori harmonization seems to mitigate this effect, it does not resolve it. It should be a focus of future research if improved MAX-DOAS a priori profiles and/or improved satellite averaging kernels can improve the situation. Based on the above, we provide an explanation in the text.

New references: EUMETSAT AC SAF operations report 1/2018, SAF/AC/FMI/OPS/RP/001, `https://acsaf.org/docs/or/AC_SAF_Operations_Report_1-2018.pdf`

Liu et al. (2019) Liu, S.; Valks, P.; Pinardi, G.; De Smedt, I.; Yu, H.; Beirle, S. and Richter, A. An improved total and tropospheric NO2 column retrieval for GOME-2 At- mospheric Measurement Techniques, 2, 2019 , 12, 1029-1057 10.5194/amt-12-1029- 2019

**Manuscript changes:** Page 31, line 1. New text. The increase in seasonal variance is caused by the interplay of the seasonal variation of the MAX-DOAS vertical profile and of the satellite vertical averaging kernel. Specifically, it is found for the Xianghe case that in wintertime averaging kernels have higher values close to the surface while MAX-DOAS NO2 profiles can also be peaked at the surface. The combination causes increased MAX-DOAS columns upon

vertical smoothing. This is also seen e.g., in the comparison of GOME-2 AC SAF GDP 4.8 NO2 product with MAX-DOAS at Xianghe (see Fig. 7.14 and 7.15 of Hovila et al. (2018) and Figs. S3 and S5 of Liu et al. (2019)). While the a priori harmonization seems to mitigate this effect, it does not resolve it. It should be a focus of future research if improved MAX-DOAS a priori profiles and/or improved satellite averaging kernels can improve the situation.

**Reviewer 1:** *Technical corrections:*

*Page 3, line 8. Define OMI here.* Done.

*Page 3, line 9. Define DOAS here.* Done.

*fig. 4. Use consistent abbreviations for stratospheric and tropospheric, i.e., strat or strato, trop or tropo.* The abbreviations in Fig. 4 have been replaced by stratospheric and tropospheric.

*Page 8, line 26. Modify by Irie et al. (e.g., 2011, Fig. 17) to by e.g., Fig. 17 in Irie et al. 2011.* Corrected.

*Page 10, line 2. AOD has been defined twice in this line, remove the 2nd one.* Corrected.

*Page 11, line 21. Change MAXDOAS to MAX-DOAS.* Corrected.

*Page 15, Figure 3. Please define MXD in the caption.* Inserted (MXD) after MAX- DOAS data.

*Page 38, line 7. Change n/a-n/a to proper page numbers.* There are several other n/a-n/a in the references. Corrected.

**Answers to reviewer 2 and corresponding changes**

**Reviewer 2:** *General Comments. The manuscript entitled Validation of Aura-OMI QA4ECV NO 2 Climate Data Records with ground-based DOAS networks: role of measurement and comparison uncertain- ties by Compernolle et al. describes the results of a validation exercise, compar- ing satellite-borne QA4ECV tropospheric and stratospheric NO 2 partial VCDs with ground-based observations from a large number of stations. The paper is very well written and repre- sents a significant contribution to the validation of satellite obser- vations. Data products, validation methodology and data screening are described in detail. Er- ror sources and potential reasons for discrepancies between ground-based and satellite-borne observations are discussed thoroughly. As far as I can judge as a non-native English speaker, there are hardly any grammatical or syntactical er- rors. I recommend the publication after addressing some minor issues as listed below. In particular, I would appreciate if the processing of the ground-based data sets and the differences between QA4ECV and bePRO data products would be discussed in some more detail.*

**Authors reply 8:** We are grateful to reviewer 2 for this positive feedback! We agree that more detail on the ground-based data sets would be useful for the reader. We give a point-by-point answer below.

**Reviewer 2:** *Specific Comments Section 2.2.2: I feel that the MAX-DOAS*

*retrieval algorithms should be described in more detail. It should be stated more clearly that the QA4ECV and the bePRO algo- rithms are distinctly different, with QA4ECD retrieving NO2 VCDs directly by dividing the dSCD from a single elevation angle by the differential AMF, while bePRO VCDs are determined by integrating a vertical NO2 profile retrieved by an OEM algorithm based on measurements from several elevation angles. There are recent studies on the performance of bePRO in comparison with other profile retrieval algorithms (Frie et al., Atmos. Meas. Tech., 2019, https://doi.org/10.5194/amt-12-2155-2019; Tirpitz et al., Atmos. Meas. Tech. Discuss, https://doi.org/10.5194/amt-2019-456) which should be cited here. I would furthermore appreciate if it would be discussed to what extent the problems with the stability of the bePRO NO2 vertical profile retrieval identified within these studies affects the quality of the data used here for OMI validation.*

**Authors reply 9:** We acknowledge that more details could have been included. We in- clude now more details on QA4ECV MAX-DOAS especially from the reference Hendrick et al. (2016) (see also our answer further below regarding the vertical grid) and on bePRO, especially from the reference Hendrick et al. (2014). We added a short text highlighting the difference between both approaches.

Furthermore, we add both suggested references. However, it is difficult to judge in how far the bePRO stability problems are important as, as stated by Friess et al. "The syn- thetic data used for the study are not necessarily representative of real measurements, especially in terms of dSCDs errors, and sensitivity tests performed by increasing the dSCDs errors but also previous publications (e.g. Hendrick et al., 2014; Vlemmix et al., 2015b) have shown that bePRO performs generally well with real measurement data, also in terms of convergence." Therefore, we rather limit ourselves to the statement that in future validation work, consideration of other retrieval algorithms, well-performing in the intercomparison exercises of Friess et al. and Tirpitz et al., would be of high interest.

**Manuscript changes:**

Page 8, line 23. New paragraph. There is a clear distinction between the QA4ECV MAX-DOAS and bePRO retrieval algorithms. In the QA4ECV MAX-DOAS algorithm, the VCD is obtained by dividing a differential SCD by a differential AMF at a single elevation angle (see section 1.3 of Hendrick et al. (2016)). In the bePRO approach (Clemer et al., 2010; Hendrick et al., 2014; Vlemmix et al., 2015) a VCD is obtained by integrating a vertical $NO_2$ profile retrieved by an OEM using measurements at several elevation angles.

Page 9, line 8. New paragraph. We note that the bePRO profile retrieval algorithm has recently been compared to several other retrieval algorithms (Friess et al. (2019), Tirpitz et al. (2020)). In future validation work, consideration of other retrieval algorithms, that perform well in the intercomparison exercises of Friess et al. (2019), Tirpitz et al. (2020) would be of high interest.

**Reviewer 2:** *P7, L10: Please provide a reference (or an URL) for the*

*description of the NDACC standard procedure.*

**Authors reply 10:** The following url was added to the manuscript: `http://ndacc-uvvis-wg.aeronomie.be/tools/NDACC_UVVIS-WG_NO2settings_v4.pdf`

**Reviewer 2:** *P8, L31: Given that QA4ECV MAX-DOAS tropospheric NO2 is deter- mined by dividing the tropospheric SCD by the tropospheric AMF, I dont understand how a vertical grid can be involved here.*

**Authors reply 11:** Details of the QA4ECV MAX-DOAS approach are available in Hendrick et al. (2016). In short, NO2 AMF is produced using the bePRO/LIDORT radiative transfer suite (Clmer et al., 2010; Spurr, 2008). The tool uses, among else, the fol- lowing input: a set of NO2 vertical profile shapes, vertical averaging kernels LUTs, geometry parameters (solar angles, viewing angles etc.), aerosol AOD vertical pro- file shapes, etc. Column AVK LUTs have been calculated based on the Eskes and Boersma (2003)s approach , using the bePRO/LIDORT RTM initialized with similar parameter values as for the AMF LUTs calculation. Interpolated AMFs, but also corre- sponding profile shapes and column averaging kernels, quantities defined on a vertical grid, are generated by the tool. We add more details of the QA4ECV MAX-DOAS approach in the text, especially on the origin of the MAX-DOAS a priori profiles and averaging kernels.

**Manuscript changes:** Page 8, line 6. New text. ”The NO2 AMF LUT are produced using the bePRO/LIDORT ...” The $NO_2$ AMF LUT are produced using the bePRO/LIDORT radiative transfer suite (Clemer et al.,2010; Spurr, 2008). This tool uses, among else, the following input: a set of NO2 verti- cal profile shapes, vertical averaging kernels LUTs, geometry parameters (like solar angles and viewing angles), aerosol AOD vertical profile shapes, etc. Col- umn AVK LUTs have been calculated based on the Eskes and Boersma (2003) approach, using the bePRO/LIDORT radiative transfer model initialized with similar parameter values as for the AMF LUTs calculation. Interpolated AMFs, but also corresponding vertical profile shapes and column averaging kernels, are generated by the tool. More detail is provided in Hendrick et al., (2016).

New reference: Spurr, R.: Light Scattering Reviews, vol. 3, chap. LIDORT and VLIDORT: Linearized pseudo-spherical scalar and vector discrete ordinat- eradiative transfer models for use in remote sensing retrieval problems, Springer, Berlin, Heidelberg, 2008.

**Reviewer 2:** *P12, L6: Explain what you mean with the term observation operator.*

**Authors reply 12:** It is already defined in the following sentence, but for better readability we rephrased that sentence.

**Manuscript changes:** Page 12, line 6. New text. This observation opera- tor is a 2-D polygon that results from the parametrization of the actual extent of the airmass to which the ZSL-DOAS measurement is sensitive.

**Reviewer 2:** *P12, L7: Which ray tracing code did you use here?*

**Authors reply 13:** UVspec/DISORT. This was added to the text.

**Manuscript changes:** Page 12, line 8. New text. Its horizontal dimensions were derived using the UVSPEC/DISORT ray tracing code (Mayer and Kylling, 2005).

**Reviewer 2:** *Last paragraph of Section 3.3 and Figure 7: It is not clear to me in which way the bias-correction for the annual mean difference has been performed - please explain in more detail.*

**Authors reply 14:** As we chose an entirely different visualization of the seasonal features in the new version of the manuscript (as requested by the other referee), this bias correction is no longer required (and thus not discussed).

**Reviewer 2:** *P23, L6: By how much has the satellite a priori profile been shifted?*

**Authors reply 15:** This depends of course on the location. We add now a range of values in the text.

p. 22, line 3. New text. The ground levels are shifted by, on average, 0.03 km (Cabauw, De Bilt) to 0.4 km (Athens, Bujumbura).

**Reviewer 2:**

*Technical Comments*

*P21, L4: characterized with → characterized by*  Corrected.

*P21, L22: Insert being before tropospheric VCD*  Inserted.

*P22, L25: Add right parenthesis: (see Eq. (1))*  Corrected.

*P23, L12 and L30: Add space between number and unit*  Corrected.

*P31, L1: does → do*  Corrected.

**Other changes**

Note: we refer here to pages and line numbers of the marked-up manuscript version (obtained using latexdiff).

p. 6, line 15. Update on total uncertainty.

p. 15. Lines 25 and following. As Figure 6 uses now a different visualization approach, the corresponding text has changed as well.

[revised manuscript text omitted]